# hDNA2 nuclease/helicase promotes centromeric DNA replication and genome stability

Zhengke Li[1], Bochao Liu[2], Weiwei Jin[1,3], Xiwei Wu[4], Mian Zhou[1], Vincent Zewen Liu[1,5,§], Ajay Goel[6], Zhiyuan Shen[2] (iD), Li Zheng[1,*] (iD) & Binghui Shen[1,**] (iD)

## Abstract

DNA2 is a nuclease/helicase that is involved in Okazaki fragment maturation, replication fork processing, and end resection of DNA double-strand breaks. Similar such helicase activity for resolving secondary structures and structure-specific nuclease activity are needed during DNA replication to process the chromosome-specific higher order repeat units present in the centromeres of human chromosomes. Here, we show that DNA2 binds preferentially to centromeric DNA. The nuclease and helicase activities of DNA2 are both essential for resolution of DNA structural obstacles to facilitate DNA replication fork movement. Loss of DNA2-mediated clean-up mechanisms impairs centromeric DNA replication and CENP-A deposition, leading to activation of the ATR DNA damage checkpoints at centromeric DNA regions and late-S/G2 cell cycle arrest. Cells that escape arrest show impaired metaphase plate formation and abnormal chromosomal segregation. Furthermore, the DNA2 inhibitor C5 mimics DNA2 knockout and synergistically kills cancer cells when combined with an ATR inhibitor. These findings provide mechanistic insights into how DNA2 supports replication of centromeric DNA and give further insights into new therapeutic strategies.

**Keywords** cell cycle arrest; centromere; centromeric DNA replication; chromosome segregation; DNA2
**Subject Categories** Cell Cycle; DNA Replication, Repair & Recombination
The EMBO Journal (2018) 37: e96729

## Introduction

The centromere is an epigenetically specified chromosomal locus in humans. It orchestrates the segregation of chromosomes during cell division and is defined by the presence of the histone H3 variant centromere protein A (CENP-A; Cleveland et al, 2003; Verdaasdonk & Bloom, 2011; McKinley & Cheeseman, 2016). The presence of the CENP-A nucleosome is sufficient to recruit the constitutive centromere-associated network and the mitotic kinetochore proteins, which are required for proper chromosome segregation (Barnhart et al, 2011; Guse et al, 2011; Mendiburo et al, 2011; Hori et al, 2013). Faithful replication of the genome, including the centromeric regions, is a challenging task; high accuracy and efficiency are needed to replicate approximately 3 billion human DNA base pairs during S-phase of the cell cycle, which lasts 6–8 h. Replication of the centromeres is even more challenging because of the presence of DNA secondary structure, and therefore likely requires specialized replication machinery. The centromeres of human chromosomes contain the largest tandem DNA family in the human genome. This family is called α-satellite DNA and has been extensively studied as a paradigm for understanding the genomic organization of tandem DNA (Schueler et al, 2001; Rudd et al, 2006). The fundamental α-satellite repeat unit consists of 171-base pair (bp) monomers, which are found in large, highly homologous arrays of up to several million base pairs at the centromeres of all human chromosomes. These tandem arrays are composed of either divergent monomers that have no detectable higher order structure or chromosome-specific higher order repeat units (HORs) characterized by distinct repeating linear arrangements of an integral set of 171-bp monomers (Rudd & Willard, 2004). This HOR structure correlates with centromere function (Schueler et al, 2001). However, the HOR structure also burdens the DNA replication machinery (Grady et al, 1992; Zhu et al, 1996; Aze et al, 2016), and requires a cellular mechanism that combines RNA/DNA helicase activity to resolve its secondary

1   Department of Cancer Genetics and Epigenetics, Beckman Research Institute, City of Hope, Duarte, CA, USA
2   Department of Radiation Oncology, Rutgers Cancer Institute of New Jersey, Rutgers Robert Wood Johnson Medical School, Rutgers, the State University of New Jersey, New Brunswick, NJ, USA
3   Department of Gastroenterology & Pancreatic Surgery, Zhejiang Provincial People's Hospital, Hangzhou, Zhejiang, China
4   Department of Molecular and Cellular Biology, Beckman Research Institute, City of Hope, Duarte, CA, USA
5   Department of Computer Science, Columbia University, New York, NY, USA
6   Center for Gastrointestinal Research, Center for Translational Genomics and Oncology, Baylor Scott and White Research Institute and Charles A. Sammons Cancer Center, Baylor University Medical Center, Dallas, TX, USA
    *Corresponding author. Tel: +1 626 301 8879; Fax: +1 626 301 8892; E-mail: lzheng@coh.org
    **Corresponding author. Tel: +1 626 301 8879; Fax: +1 626 301 8892; E-mail: bshen@coh.org
    §Correction added online on 23 May 2018 after first online publication: the author name has been corrected.

structure and structure-specific nuclease activity, such as that found in DNA2, to process DNA replication intermediates.

Mechanistically, it is currently unknown how centromeric DNA is replicated. It is unlikely that during DNA replication general nucleases such as flap endonuclease 1 (FEN1) can process structured DNA sequences, such as those found in telomeres and centromeres (Henricksen *et al*, 2000; Tarantino *et al*, 2015). In contrast to FEN1, DNA2 has both helicase and endonuclease activities (Budd *et al*, 1995), which could work together to resolve these challenging DNA structures. The enzymatic activities of DNA2, which reside in a RecB-like nuclease domain, target single-stranded DNA (ssDNA; Bae *et al*, 1998), DNA flaps (Kao *et al*, 2004; Copeland & Longley, 2008; Stewart *et al*, 2010), and DNA secondary structures (Lee *et al*, 2013; Lin *et al*, 2013). The C-terminal superfamily 1 helicase domain of DNA2 can unwind kilobases of dsDNA from the 5′ end *in vitro* (Pinto *et al*, 2016). However, the biological functions of the helicase activity of DNA2 in cells are unknown. Genetic inactivation of either the helicase or nuclease activity of DNA2 in cells from a wide range of organisms, including yeast and humans, induces permanent cell cycle arrest (Budd & Campbell, 1997; Budd *et al*, 2000; Lee *et al*, 2000; Zheng *et al*, 2008; Duxin *et al*, 2009, 2012). In yeast, the cell death has been ascribed to the role of yeast Dna2 in processing DNA replication intermediates (Budd *et al*, 1995; Kang *et al*, 2000, 2010; Olmezer *et al*, 2016). In humans, DNA2-depleted cells undergo arrest during late-S/G2 phase of the cell cycle (Duxin *et al*, 2009, 2012). However, our understanding of the role of DNA2 in double-strand break (DSB) end resection cannot explain why DNA2 is required for viability in unperturbed cells (Zhu *et al*, 2008; Niu *et al*, 2009; Cejka *et al*, 2010; Chen *et al*, 2011), and indeed, the role of DNA2 during replication has yet to be defined.

In the current study, we demonstrate that DNA2 predominantly binds to centromeric DNA regions. Single-molecule analysis of replicated DNA (SMARD) revealed that loss of DNA2 results in stalled replication of centromeres. These centromeric DNA replication defects led to activation of the DNA damage checkpoint kinase ataxia telangiectasia and Rad3 related (ATR), which contributes to cell cycle arrest in late-S/G2 phase. DNA2 nuclease- or helicase-deficient cells showed compromised loading of CENP-A onto chromatin and loss of intact centromeric DNA. In addition, cells that escaped the G2/M checkpoint showed inappropriate formation of the metaphase plate and chromosomal mis-segregation. Inhibition of both DNA2 and ATR had synergistic activity for killing breast, colorectal, and non-small-cell lung cancer cells. Collectively, these studies support a model wherein the concerted action of DNA2 helicase and nuclease activity is crucial to centromeric DNA replication.

# Results

## DNA2 binds preferentially to centromeric DNA and is required for centromeric DNA replication

To explore the function of DNA2 in DNA replication, we first globally mapped the localization of DNA2 on chromatin. We used chromatin immunoprecipitation (ChIP) with a DNA2 antibody to pull down DNA2-associated chromatin in non-synchronized HCT-116 cells and then conducted whole-genome DNA sequencing. We found that DNA2 predominantly bound to the centromeric α-satellite regions (Figs 1A columns 1–3, and EV1A). This result was validated by using qPCR to compare the fold enrichment over input for the centromeric and non-centromeric regions (Fig 1B). The centromeric regions contained 58% of the DNA2-associated DNA, representing a 33.5-fold enrichment (Fig 1C). This finding supports our hypothesis that DNA2 is important in replication of centromeric DNA. In contrast to DNA2, when we used FEN1 ChIP DNA as a template, qPCR did not show any preferential recruitment of FEN1 to the centromeric DNA (Figs 1B and EV1B), suggesting these cellular nucleases have differing functions during DNA replication.

Based on the preferential binding of DNA2 to centromeric DNA, we hypothesized that introducing a deficiency in DNA2 would retard centromeric DNA replication. To test this, we used HCT-116 cells engineered for Cre-mediated excision of DNA2 (DNA2$^{Flox/−/−}$) in response to treatment with 4-hydroxytamoxifen (4-OHT, creating DNA2-null cells) and vehicle-treated cells (Karanja *et al*, 2014; Thangavel *et al*, 2015). We labeled newly synthesized DNA with the synthetic thymidine analog bromodeoxyuridine (BrdU) for 32 h, corresponding to at least one DNA2$^{Flox/−/−}$ cell cycle. The BrdU-incorporated nascent chromatin was then depleted using a BrdU antibody, leaving behind the BrdU-negative, under-replicated DNA. In 4-OHT-treated DNA2-null cells, approximately 10% of the genomic DNA was BrdU-negative, under-replicated DNA, but there was no detectable under-replicated DNA in the vehicle-treated cells (Fig EV1C). This suggests that DNA2 is required to complete DNA replication.

To define the under-replicated DNA regions in DNA2-null cells, we conducted whole-genome DNA sequencing of the under-replicated DNA. After normalization to genomic input DNA from the same cells, 12.9% of the peaks from the under-replicated DNA aligned with the centromeric DNA regions (Fig 1A column 4, and C), representing an 8.5-fold enrichment (Fig 1C). In addition, among the peaks that overlapped between the DNA2 pull-down and under-replicated regions, two-thirds fell into the centromeric regions, representing a 48-fold enrichment (Fig 1A column 5, and C). We

**Figure 1. DNA2 binds preferentially to centromeric DNA and is required for centromeric DNA replication.**

A   DNA peaks were identified as described in Materials and Methods. Shown is an alignment of whole-genome sequencing results to the Hg38 genome. Column 1, Hg38 genome structure (UCSC genome browser); column 2, Hg38 centromere locations; column 3, DNA2 binding sites in HCT-116 cells identified from ChIP-seq; column 4, BrdU-negative DNA from DNA2-null cells; column 5, common peaks among DNA samples shown in lanes 3 and 4. The chromosome number is listed on the left of the plot.

B   Fold enrichment of centromeric versus non-centromeric DNA over input as determined by qPCR of DNA2 and FEN1 ChIP samples. Shown by chromosome (Chr) are the means ± SDs of three biological repeats. See Table EV1 for primers used.

C   Quantitative analysis of the DNA peaks and bases shown in panel (A).

D   The most highly enriched repetitive centromeric/paracentromeric motif is shown. Letter size reflects its frequency at the position. *P*-value of 1.2e-88 derived by Fisher's exact test.

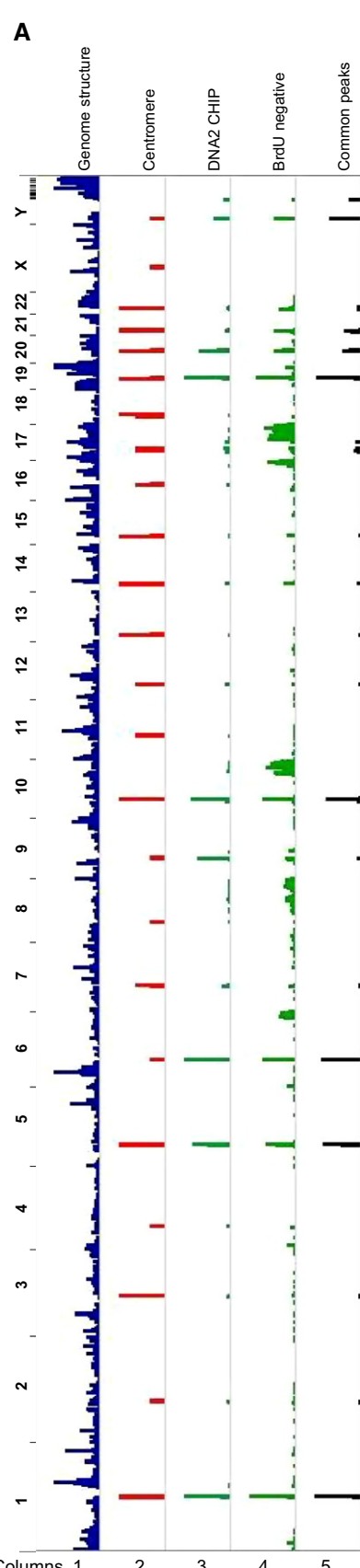

**A**

**B**

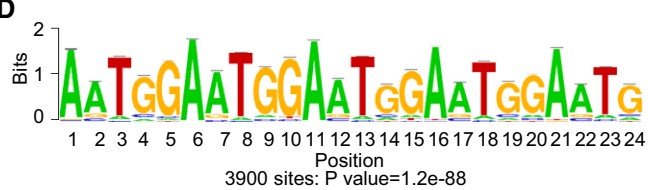

Chromosome number (Y to 1)

Centromeric /non-centromeric DNA levels

■ Anti-DNA2 ChIP
□ anti-FEN1 ChIP

**C**

DNA2 binds to centromeric DNA and is required for centromeric DNA integrity

| Pull-downs | Sequencing peaks | Peaks overlap with centromere (%) | Bases of peaks in centromere | Bases of peaks not in centromere | Fold enrichment in centromere[2] |
|---|---|---|---|---|---|
| DNA2 | 2260 | 1311 (58.0) | 341530 | 539645 | 33.5 |
| BrdU[1] | 13453 | 1736 (12.9) | 427920 | 2659492 | 8.5 |
| DNA2/BrdU[1] | 502 | 335 (66.7) | 68711 | 76265 | 47.6 |

[1]BrdU negative. [2]Base numbers in the sequencing peaks in the reference genome: Centromere: 59,546,786; Entire genome: 3,209,286,105; % centromere: 1.86%

**D**

3900 sites: P value=1.2e-88

**Figure 1.**

then computed the motifs derived from the common peaks. The number and percentage of peaks associated with each motif, the mean number of sites per peak, and the *P*-value for each motif are shown in Fig EV2. The most frequent motif was (TGGAA)n (Figs 1D and EV2), which is often found in centromeric DNA regions (Grady *et al*, 1992; Catasti *et al*, 1994; Kipling *et al*, 1995; Zhu *et al*, 1996; Barry *et al*, 1999). Thus, our data suggest that DNA2 preferentially localizes to the centromere and is required for centromeric DNA replication.

## Concerted action of the nuclease and helicase activities of DNA2 facilitates processing of intermediate structures during centromeric DNA replication

During centromeric DNA replication, DNA secondary structures can arise from the single-stranded template DNA, which may block replication fork progression in cells. We analyzed the ability of human wild-type (WT) DNA2 (hDNA2), as well as D294A nuclease-deficient (ND) and K671E helicase-deficient (HD) hDNA2 mutant proteins (Masuda-Sasa *et al*, 2006; Lin *et al*, 2013) to cleave the DNA replication intermediates that formed at various regions of the genome. Figure 2A–C illustrates the DNA structures, and Fig 2D and E shows cleavage of 5′- and 3′-labeled substrates, respectively. We first confirmed the activity of WT hDNA2 and the hDNA2 mutant proteins using an established DNA flap DNA substrate (Lin *et al*, 2013). Consistent with previous reports (Masuda-Sasa *et al*, 2006; Lin *et al*, 2013), both WT and HD hDNA2, but not ND hDNA2, efficiently cleaved a typical flap ssDNA that lacked repetitive DNA sequences (Fig 2A, D, and E, lanes 2–11), sequentially cleaving ~ 10 nucleotides (nts) from the 5′ end. When incubated with a model DNA substrate mimicking the hairpin DNA structure generated by centromeric repetitive DNA, i.e., (TGGAA)$_6$ (Fig 2B), WT hDNA2, but not the ND or HD mutants, effectively separated the DNA helix and cleaved the DNA structure, also ~ 10 nts from the 5′ end, in a sequential manner (Fig 2D and E, lanes 12–21). Because the highly enriched centromeric 171-nt α-satellite DNA that we found in our CHIP-seq experiments (Waye & Willard, 1986; Bloom, 2014) can form stable longer stem-loop structures (Aze *et al*, 2016), we designed a third DNA substrate that mimicked this structure (Fig 2C). We found that WT hDNA2, but not the ND or HD mutants, could effectively cleave the stem-loop structure (Fig 2D and E, lanes 22–31). Importantly, the combined helicase and nuclease activities of DNA2 were required to separate the 5′ complementary DNA, which then became ssDNA that was cleavable by DNA2 nuclease at cleavage sites clustered ~ 10 nts from the 5′ end of the single-stranded DNA (Fig 2C). We also designed additional DNA structures (Fig EV3), which contained one, two, or three stem/loop structures, and comprehensively mapped the cleavage sites based

on the product sizes shown in the gel images. Our cleavage site mapping indicated that DNA2 nuclease activity only works on the ssDNA regions and requires its helicase activity, which is similar as previous demonstrations (Lin *et al*, 2013; Ronchi *et al*, 2013; Liu *et al*, 2016). This result supports the following model for resolution of multiple stem-loop structures during centromeric DNA replication: The DNA2 helicase activity first separates the stem to make a single ssDNA that is long enough (~ 10 nts) for nuclease cleavage; then, when the ssDNA at the junction is exposed, DNA2 cleaves the whole hairpin structure and removes the structured DNA at the replication fork.

## DNA2 is required for centromeric DNA replication fork initiation and progression

To assess DNA replication activities at the centromeric region, we modified the SMARD assay that was previously established for studying telomere replication (Norio & Schildkraut, 2001; Drosopoulos *et al*, 2015). In the SMARD assay, the cells were labeled with iodo-deoxyuridine (IdU) for 4 h during a first labeling period, followed by labeling with chloro-deoxyuridine (CldU) for another 4 h in a second labeling period (Norio & Schildkraut, 2001; Demczuk *et al*, 2012; Drosopoulos *et al*, 2015). These 4-h labeling periods provided sufficient time to replicate the DNA regions analyzed, so that we could use the differential labeling, which represented different DNA replication stages, to analyze the initiation and progression of DNA replication forks across the genome (Norio & Schildkraut, 2001). Such a relatively long labeling period is particularly important for analyzing the replication of difficult-to-replicate DNA regions using the SMARD assay (Drosopoulos *et al*, 2015). Following this labeling protocol (Norio & Schildkraut, 2001; Demczuk *et al*, 2012; Drosopoulos *et al*, 2015), we then combed genomic DNA from these cells onto coverslips and used centromere-specific fluorescent DNA probes to identify centromeric DNA.

Only DNA fibers that had IdU labeling and/or CldU labeling at non-centromeric or centromeric regions were analyzed to exclude including of DNA fibers from non-replicating cells. Presence of either red- or green-only tracks was classified as initiation events, while the length and distribution of the green segment within red-green tracks were evaluated to reflect the rate of replication progression. We found that the numbers of tracks with either red- or green-only and the lengths and distributions of green segments within red-green tracks at non-centromere regions in DNA2 mutant cells (null, ND, and HD) were similar to those in WT cells (Fig 3A and B), which agreed with a previous report (Thangavel *et al*, 2015). However, the numbers of tracks with only either red or green, and the lengths and distributions of green segments within red-green tracks at centromeric regions in DNA2 mutant cells were

---

**Figure 2.  DNA2 helicase and nuclease activities are required for removal of *in vitro* centromeric DNA secondary structures.**

A–C  Panel (A) shows flap DNA structure (lanes 2–11 in panels D and E). Panel (B) shows the (TGGAA)n motif structure (lanes 12–21 in panels D and E). Panel (C) shows α-satellite DNA structure (lanes 22–31 in panels D and E). Red arrows mark the cleavage sites.

D, E  5′-radiolabeled (panel D) or 3′-radiolabeled (panel E) non-centromeric DNA substrates (lanes 2–11) or centromeric substrates (lanes 12–31) were incubated with purified DNA2 for 5, 10, or 20 min. Representative images from at least three biological repeats are shown. The DNA2 cleavage signatures are shown in panels (A–C), along with a model that illustrates the resolution of DNA secondary structure, as predicted by the "RNAfold" software package. Resolving of the DNA substrates required different amounts of DNA2 protein: 0.5 ng for the DNA flap, 10 ng for (TGGAA)n, and 7.5 ng for the α-satellite stem-loop structure.

Source data are available online for this figure.

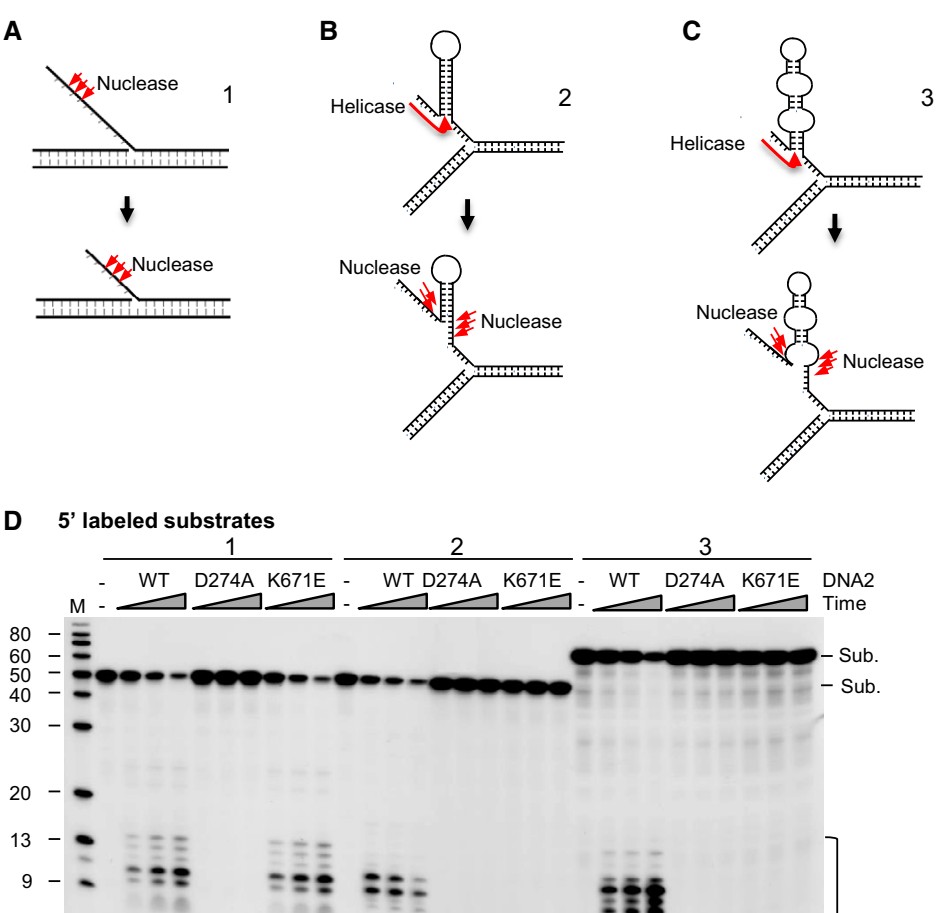

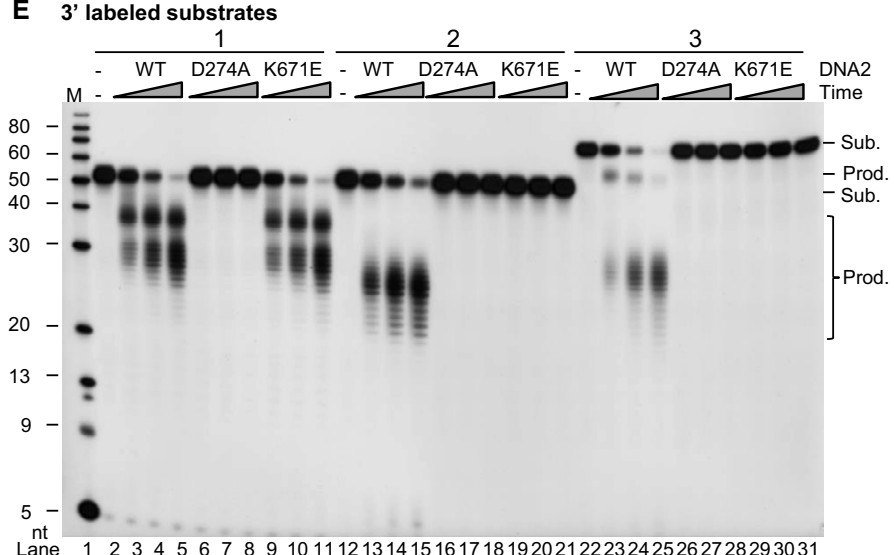

**Figure 2.**

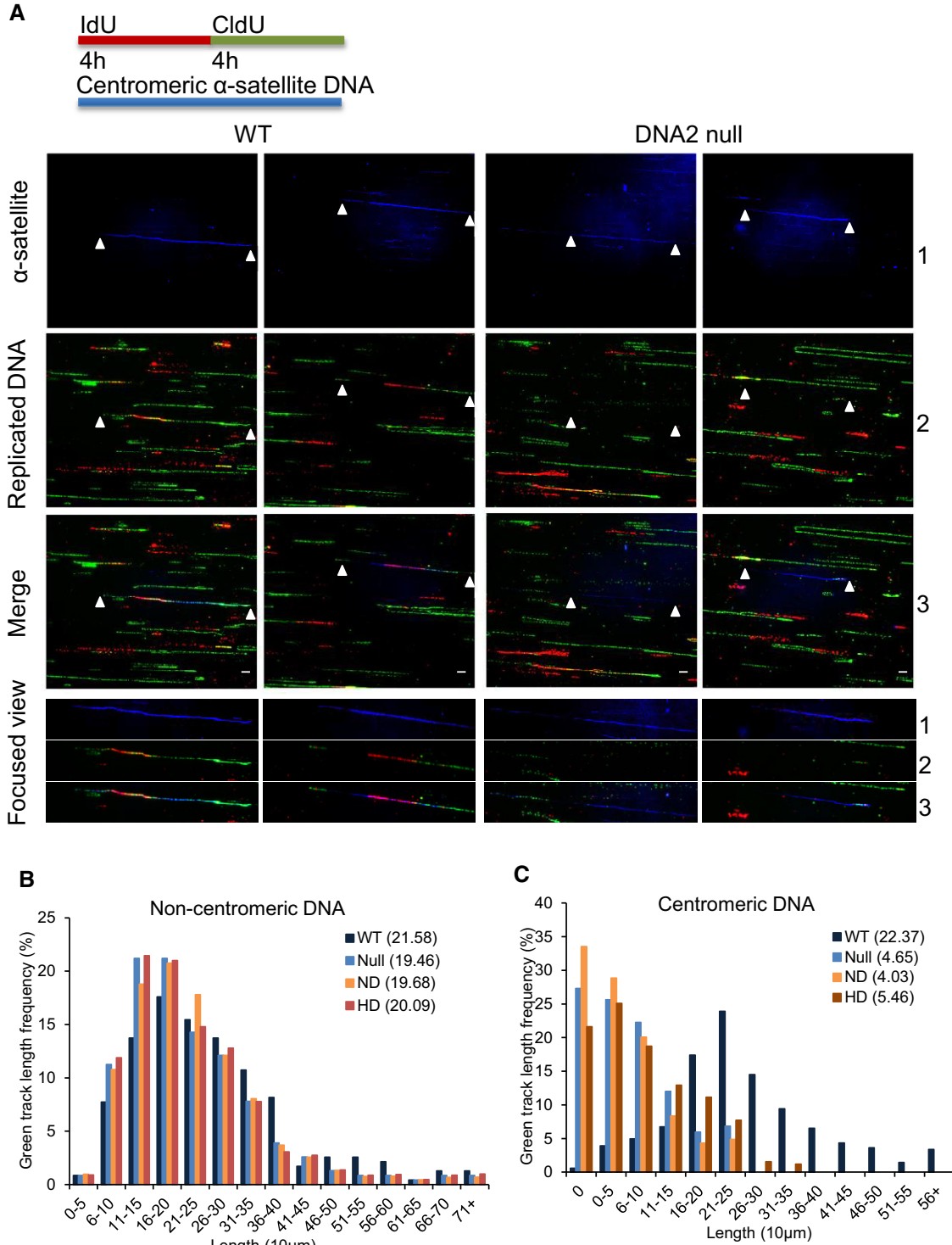

**Figure 3.  DNA2-mutant cells show reduced replication initiation and slower centromeric replication fork progression assessed by single-molecule analysis of replicated DNA (SMARD).**

A    DNA2$^{Flox/-/-}$ cells that stably expressed DNA2-WT, DNA2-ND, and DNA2-HD were treated with 1 μM 4-OHT for 72 h to remove endogenous DNA2 and then cultured with IdU (40 μM) for 4 h followed by CldU (200 μM) for 4 h. Shown are representative images of combined centromere-specific DNA and surrounding non-centromeric DNA. Bottom panels show a focused view of the centromeric DNA regions. Scale bars, 10 μm.

B, C    Quantification of the length and frequency of green tracks in the non-centromeric (B) and centromeric (C) DNA regions (out of three independent replicates, $n \geq 200$ tracks were scored for each dataset). To exclude counting of compromised replication initiation (no any countable tracks) that could be caused by changes in cell cycle distribution, the fibers with centromeric DNA that had at least either red or green tracks were counted to ensure the cell was in S-phase. Median track lengths are indicated in parentheses.

both remarkably reduced (Fig 3A and C). Twenty to thirty percent of DNA fibers displayed no detectable green or red tracks at centromeric regions in the DNA2 null, ND, or HD mutant cells as compared to only ~ 1% of fibers for WT cells (Fig 3A and C), indicating DNA replication initiation defects at centromere regions in mutant cells. With respect to red-green tracks, the median lengths of green segments at centromeric regions in DNA2 null, ND, and HD cells were 46.5, 40.3, and 54.6 μm, respectively, compared to a median length of 223.7 μm for centromeric green segments in WT cells (Fig 3C). Approximately 25% of these green tracks within centromeric DNA in DNA2-null and ND or HD mutant cells were 0–50 μm, and < 5% of the green tracks were longer than 210 μm. In contrast, < 5% of centromeric green tracks in WT cells were 0–50 μm, and approximately 70% of the green tracks were longer than 210 μm. These findings indicated defects in DNA replication initiation and replication fork progression at the centromeric regions in DNA2 mutant cells.

### Both the nuclease and helicase enzymatic activities of DNA2 are required for centromeric DNA integrity

Because our results suggested that DNA2 deficiency impairs centromeric DNA replication, we assessed centromeric DNA replication defects in DNA2-null cells with and without complementation with WT, ND, and HD DNA2. Using qPCR, we detected reduced abundance of intact centromeric DNA in DNA2-null cells as compared to vehicle-treated cells (Fig 4A). We also found a significant loss of intact centromeric DNA, which was rescued by WT DNA2, but not by the ND and HD DNA2 mutants (Fig 4A). This suggests that the DNA2 helicase or nuclease activity, or both, is critical for centromere maintenance.

Centromeres of complex eukaryotes are defined by the presence of a centromere-specific histone H3 variant, CENP-A (Verdaasdonk & Bloom, 2011; Bloom, 2014; McKinley & Cheeseman, 2016). Although CENP-A is deposited on the centromere during early G1 phase (Dunleavy et al, 2009; Foltz et al, 2009), the chromatin loading of CENP-A may reflect the intactness of the centromeric DNA (Bodor et al, 2014; Aze et al, 2016). In addition, the presence of the CENP-A nucleosome is sufficient to recruit the constitutive centromere-associated network and mitotic kinetochore proteins, which are required for proper chromosome segregation (Barnhart et al, 2011; Guse et al, 2011; Mendiburo et al, 2011; Hori et al, 2013), and thus define the centromere (Cleveland et al, 2003; Verdaasdonk & Bloom, 2011; McKinley & Cheeseman, 2016). For this reason, we analyzed CENP-A in the absence of DNA2, and observed decreased chromatin loading of CENP-A (Fig 4B and C). This under-recruitment of CENP-A was not due to cell cycle effects (Fig EV4A–C), nor was it because of the mismatch repair (MMR)-deficient status of HCT-116 cells (Fig EV4D). Similar amounts of CENP-A were found to be bound to chromatin in cells at S/G2 phase (Foltz et al, 2009; Fig EV4A–C) when comparison was made between cells treated with or without nocodazole or RO-3306 or between MMR-proficient and MMR-deficient cells (Fig EV4D). Therefore, both the reduced chromatin loading of CENP-A and incomplete replication of centromeric DNA in DNA2-null cells suggest the subsequent loss of centromere function. In addition, levels of chromatin-bound CENP-A in DNA2 null cells were restored to those in control cells by complementation with WT DNA2 but not with ND or HD DNA2 (Fig 4B

and C). Overall, these results suggest that the concerted action of the DNA2 helicase and nuclease activities is crucial for centromeric DNA replication in cells.

### DNA2 deficiency causes failure of mitotic entry, centromere abnormalities, and dysfunctional chromosomal segregation

To determine whether DNA2 deficiency arrests cell cycle progression, we compared the mitotic entry of WT and DNA2-null cells, using phospho-histone H3 (s10) as a mitotic cell marker. The number of mitotic cells was visualized by fluorescence microscopy (Fig 5A and B) and quantified by flow cytometry (Fig 5C and D). Fewer DNA2-null than WT cells entered mitosis, both with and without the addition of nocodazole (100 μg/ml) to enrich for mitotic cells (Fig 5A–D). Furthermore, more DNA2-null cells as compared to WT cells accumulated in late-S/G2 phase (Fig EV4A and B), suggesting that DNA2 is required for progression through late-S/G2 phase to mitosis, which is consistent with previous reports (Duxin et al, 2009, 2012).

A small portion of DNA2-null cells spontaneously escaped cell cycle arrest and entered mitosis (Fig 5A–D). We hypothesized that such mitotic cells might have defects in chromosomal segregation due to improper centromere DNA replication. To test this, we examined the attachment of microtubules to the centromeres during metaphase. We found that the chromosomal centromeres in cells lacking DNA2 were not attached to the spindle fibers nor was the metaphase plate appropriately formed (Figs 5E lower right, and EV5A). The chromatin bodies were disassociated from the metaphase plate, as indicated by the appearance of chromosomal misalignment that lacked calcium-responsive transactivator (CREST) or CENP-A immunoreactivity, and were not attached to α-tubulin (Figs 5E white arrows, and EV5A movie). Misaligned chromosomes were found in 68% of the DNA2-null cells as compared to in 6% or fewer of the uninduced or WT cells (Figs 5F and EV5B). In addition, the chromosome congression index (chromosome length/width) decreased from 1.98 in uninduced DNA2$^{Flox/-/-}$ cells to 1.25 after 4-OHT induction (Figs 5G and EV5C). Consistent with these results, 35–40% of the DNA2$^{Flox/-/-}$ cells showed chromosome segregation abnormalities with lagging chromosomes as compared to 10% of the WT cells (Fig EV5D and E). Taken together, these data suggest that cells lacking DNA2 lose centromere integrity and are unable to segregate chromosomes appropriately. The loss of mitotic entry and subsequent cell death (Fig EV4E) occurred because of cell cycle arrest in late-S/G2 phase, which is consistent with previous observations (Duxin et al, 2009, 2012).

### Compromised centromeric DNA replication induces replication stress checkpoint signaling and late-S/G2 phase arrest

Autophosphorylation of ATR on Thr1989 (phospho-ATR) is considered a marker of ATR activation, and its recruitment to RPA-ssDNA is a molecular switch for launching a robust checkpoint response (Liu et al, 2011; Nam et al, 2011). To determine whether DNA2 deficiency resulted in ATR activation across the genome, we performed ChIP-seq analysis of the recruitment of phospho-ATR to chromatin of DNA2-null cells. From a whole-genome perspective, we found peaks of DNA at centromeric

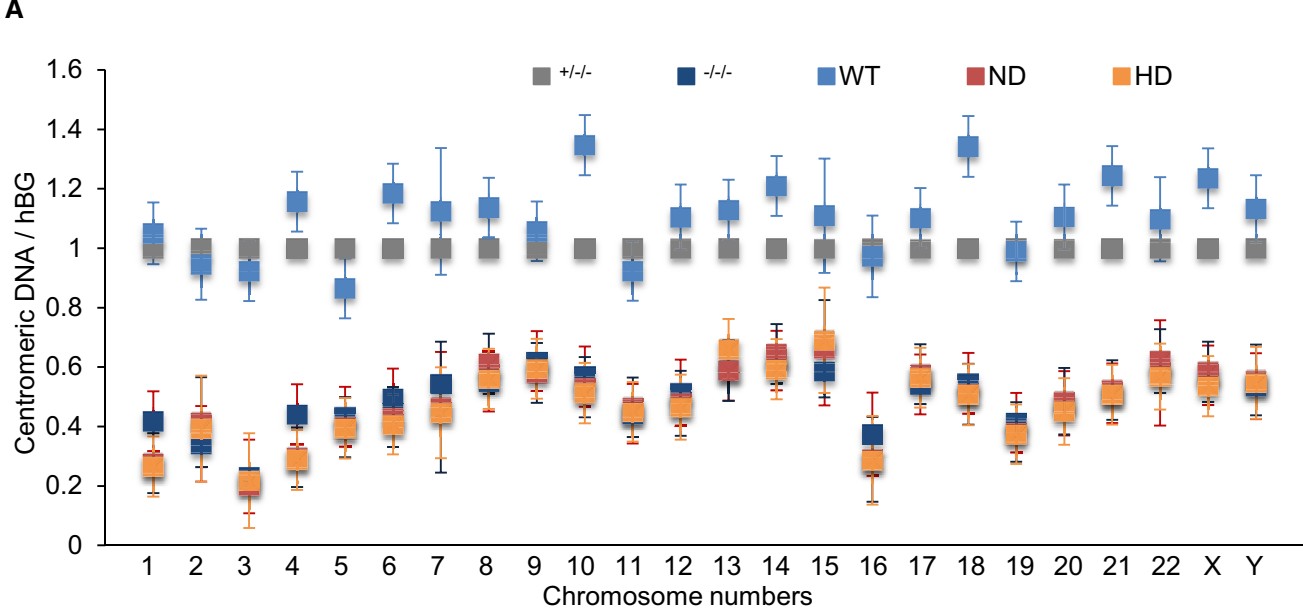

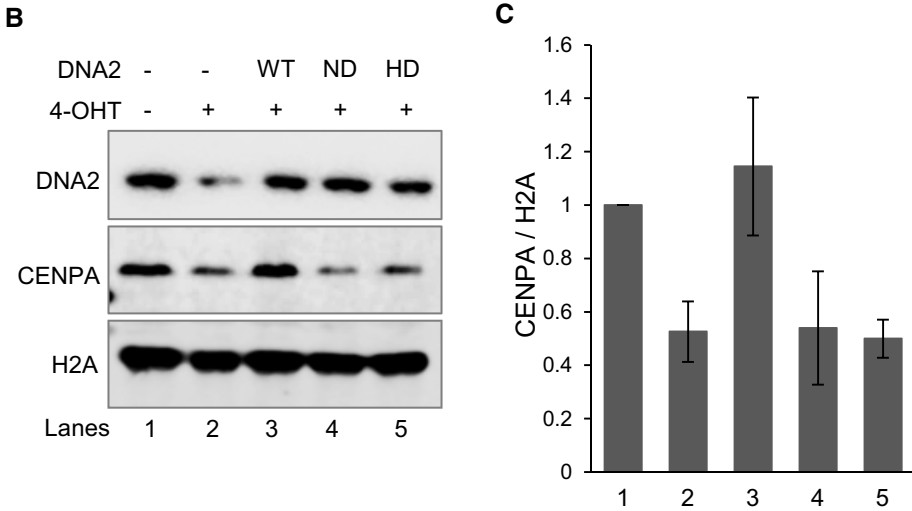

**Figure 4. Conditional knockout of DNA2 or inactivation of its nuclease/helicase activity compromises centromere integrity.**

A    DNA2$^{Flox/-/-}$ cells were transfected with DNA2-WT, DNA2-ND, and DNA2-HD constructs, selected with 250 μg/ml hygromycin B for 2 weeks, and then treated with 1 μM 4-OHT for 72 h to remove endogenous DNA2. The presence of intact centromeric DNA was quantified by qPCR from the extracted genomic DNA. Shown by chromosome (Chr) are the means ± SDs of three biological repeats after normalization to the hGB gene.

B, C    Western blot analysis (left) of DNA2 and chromatin-bound histones in cell extracts from the same cells as in panel (A). Quantification of the mean ± SD of the densitometry of the bands from four replicates is shown in panel (C). Lanes: 1, DNA2$^{Flox/-/-}$ cells; 2, DNA2$^{-/-/-}$, 3, DNA2-WT; 4, DNA2-ND; 5, DNA2-HD.

Source data are available online for this figure.

regions bound to phospho-ATR in DNA2-null cells (Fig 6A, column 4, and comparison between columns 4 and 2). Quantitatively, compared with cells that expressed WT DNA2, this represented a 4.9-fold enrichment of the percentage of DNA peaks that were located in centromeric regions (Fig 6B). At the cellular level, we consistently observed that DNA2 knockout led to co-localization of phospho-ATR and the FISH signal for centromeric DNA (Fig 6C). The activation of ATR may result from the inability of

DNA2 to process stalled forks (Budd & Campbell, 1995; Budd et al, 1995; Hu et al, 2012; Thangavel et al, 2015; Bass et al, 2016) by resolving replication intermediates arising at least in part from centromeric DNA (Fig 1A columns 2 and 4, and Figs 2 and 3), and/or defects in CENP-A deposition, which is critical for suppressing ATR activation (Aze et al, 2016). We also observed accumulation of RPA foci at centromere sites in DNA2 null cells but not in WT cells (Fig 6D), nor in DNA polymerase inhibitor

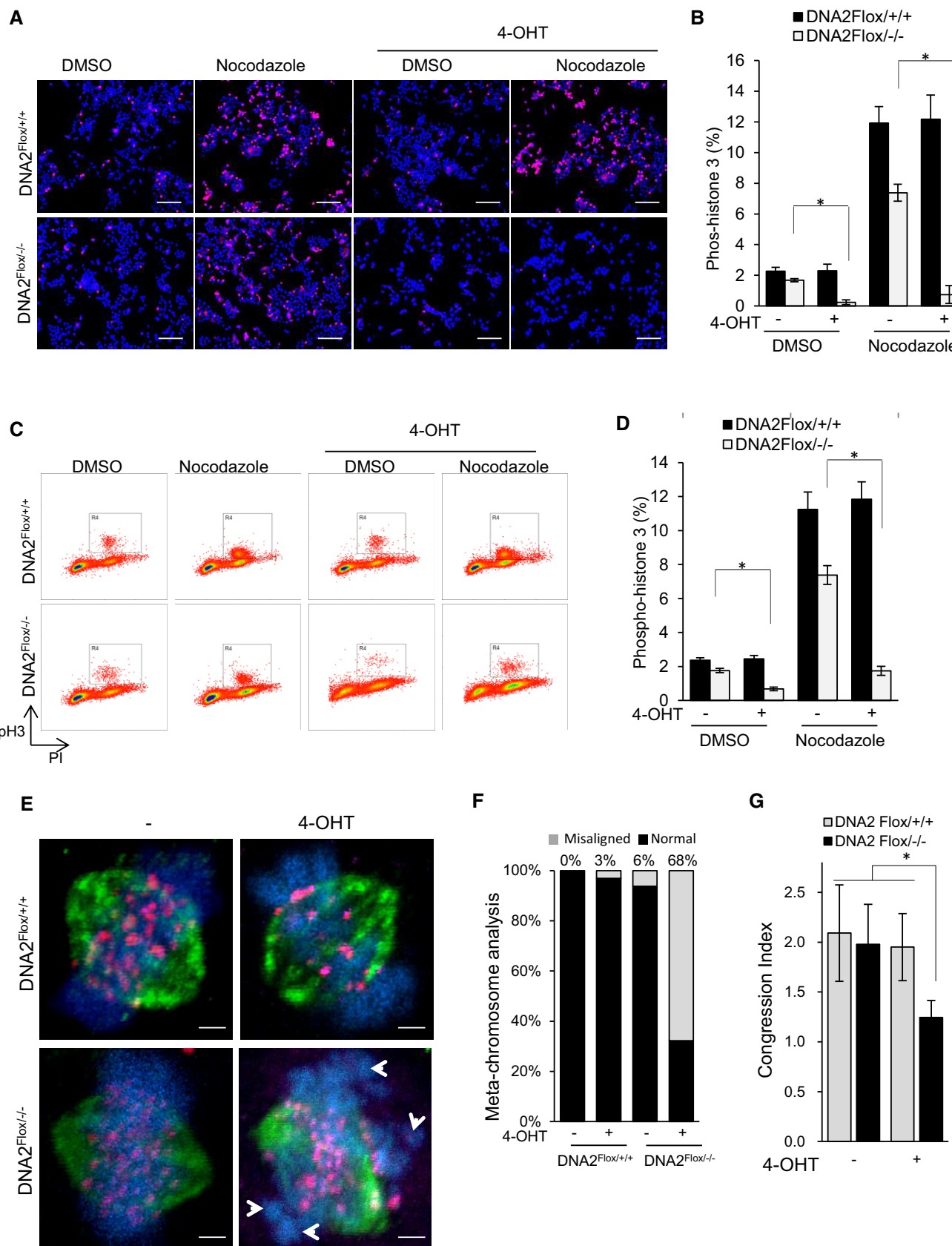

**Figure 5.**

**Figure 5.    DNA2 deficiency causes centromeric abnormalities and dysfunctional chromosomal segregation.**

A, B    DNA2$^{Flox/+/+}$ and DNA2$^{Flox/−/−}$ cells were treated with or without 1 μM 4-OHT for 48 h, then treated with or without 100 μg/ml nocodazole for 6 h. Cells were then stained with phospho-histone H3 (red) and DAPI (blue). Representative immunofluorescence micrographs of mitotic cells are shown. Scale bars, 100 μm. Panel (B) shows mean ± SD on percentage of cells positive for phospho-histone H3 from three biological repeats. Unpaired two-tailed t-test was performed to calculate a P-value (*P < 0.05).

C, D    The same cells as in panel (A) were analyzed by flow cytometry to count the number of mitotic cells. Phospho-histone H3 (y-axis) and PI (x-axis). Panel (D) shows the percentage of phospho-histone H3-positive cells (mean ± SD of four biological repeats) as in (C). *P-value is < 0.05 using unpaired two-tailed t-test.

E    Following sequential treatment with thymidine and RO-3306, synchronized cells (at the G2/M border) were released to metaphase. Representative micrographs of cells that entered metaphase are shown. Cells were stained with DAPI (DNA counterstain, blue), anti-tubulin (green), anti-CENP-A (pink), and anti-CREST (red). Arrowheads indicate chromosomal regions lacking CREST and CENP-A immunoreactivity. Scale bars, 2 μm.

F    Percentages of misaligned chromosomes, defined as a cluster of DNA lacking CREST and CENP-A, and lacking attachment to tubulin (mean ± SD from ∼ 50 cells). See also Fig EV5B.

G    Congression index (length/width) of DAPI-labeled DNA (mean ± SD from ∼ 50 cells). See also Fig EV5C. *P-value is < 0.05 by one-way ANOVA followed by unpaired two-tailed t-test.

aphidicolin (APH)-treated WT cells, which agreed with previous report that APH does not induce ATR activation and RPA accumulation at centromere (Aze *et al*, 2016). Our data suggest that RPA bound to the ssDNA region at the centromere for ATR activation in DNA2 null cells.

**The DNA2 inhibitor C5 has a synergistic inhibitory effect with ATR inhibition**

ATR activation at stalled replication forks was previously shown to protect cells from replication catastrophe and cell death (Buisson *et al*, 2015). Thus, modulation of ATR activation activity is important for ensuring sufficient time for the DNA replication machinery to replicate the centromere (Aze *et al*, 2016). Based on the known role of DNA2 in centromere replication, we hypothesized that ATR activation is critical for protecting cells from deleterious effects arising from DNA2 deficiency-induced replication stalling at centromeres and other difficult-to-replicate regions. Therefore, simultaneous inhibition of DNA2 and ATR could have an additive or synergistic effect in killing cancer cells. To test this hypothesis, we inhibited DNA2 with the specific DNA2 inhibitor C5 (Liu *et al*, 2016) and/or blocked ATR activation with the ATR inhibitor VE-821 in several commonly used aggressive cancers cell lines such as MCF7 (breast cancer; Fig 7A–C), and HCT-116 (colorectal cancer) and H460 (non-small-cell lung cancer; Fig 7D and E). The combination of C5 with VE-821 synergistically killed MCF7, HCT-116, and H460 cells (Fig 7B–E). The combination index for VE-821 (0.2 μM) and C5 (1.0 μM) was 0.19 in MCF7 cells, 0.22 in HCT-116 cells [VE-821 (0.2 μM) and C5 (8.0 μM)], and 0.29 in H460 cells [VE-821 (0.2 μM) and C5 (10.0 μM)], indicating a strong synergy between two drugs that effectively target different types of cancer cells. At a mechanistic level, we showed that chemical inhibition of DNA2,

similar to genetic knockout of DNA2, leads to cell cycle arrest at late-S/G2 phase and failure of mitotic entry (Fig 7F and G). Combined chemical inhibition of DNA2 and ATR had a synergetic effect in arresting the cells at S/G2 cell phase, reducing their mitotic entry, and causing apoptosis (Fig 7F–I). In contrast, we did not observe such synergy with combined use of inhibitors of DNA2 and ATM (Fig EV6A–C).

# Discussion

It is well known that α-satellite DNA in centromere regions is subject to secondary structures when the DNA replication machinery opens double-stranded DNA. These are difficult-to-replicate regions with secondary structures that are obstacles for progression of the DNA replication fork. Indeed, the DNA secondary structure within centromeric DNA mimics replication challenges induced by replication stressing reagents and may require involvement of DNA repair factors such as MSH2-6, XRCC5, MUS81, XRCC1, and RAD50 (Aze *et al*, 2016). However, because centromeric DNA accounts for < 5% of the genome, the impact of the secondary structures specifically on these regions can be easily overlooked by standard DNA fiber analyses. Therefore, in this study, we modified the SMARD technology to specifically examine the DNA replication dynamics in centromere regions. We provide evidence that DNA2 maintains genome stability by facilitating replication of centromeric DNA. We discovered that the DNA2 nuclease/helicase predominantly localizes to the centromeric α-satellite regions of the human nuclear genome (Fig 1) and is required for completion of DNA replication through the centromeric DNA (Fig 3). In DNA2-null cells, the under-replicated centromeric DNA impairs chromosome segregation (Fig 5), leading to the activation and the enrichment of ATR checkpoint kinase at centromeric

**Figure 6.    Stalled replication signaling is detected at centromeric DNA in DNA2 null cells.**

A    ChIP-pATR peaks from WT and DNA2-null samples were identified as described in Materials and Methods. Shown is an alignment of whole-genome sequencing results to the Hg38 genome. Column 1, Hg38 genome structure (UCSC genome browser); column 2, Hg38 centromere locations; column 3, phospho-ATR binding sites in HCT-116 cells identified from ChIP-seq; column 4, phospho-ATR binding sites in the DNA2-null cells; column 5, common peaks of DNA samples in lanes 3 and 4; column 6, phospho-ATR peaks that were identified in the DNA2-null cells only. The chromosome number is listed on the left of the plot.

B    Quantification analysis of the alignment of DNA peaks (shown in A) to different repetitive DNAs. Repetitive regions in the human genome (hg38) were downloaded from the UCSC genome browser "RepeatMasker" track.

C    Representative images of IF-FISH analysis of phospho-ATR (red) and CENP-B box (green) that indicate centromere locations after cells were incubated for 72 h with 1 μM 4-OHT. As a comparison analysis, WT cells were stained after incubation with aphidicolin (APH, 5 μg/ml) for 2 h. Arrow marks co-localization. Scale bars, 2 μm.

D    Representative images of cells that were treated as described in (C) were stained with RPA (red) and CENPB box (green). Arrow marks co-localization. Scale bars, 2 μm.

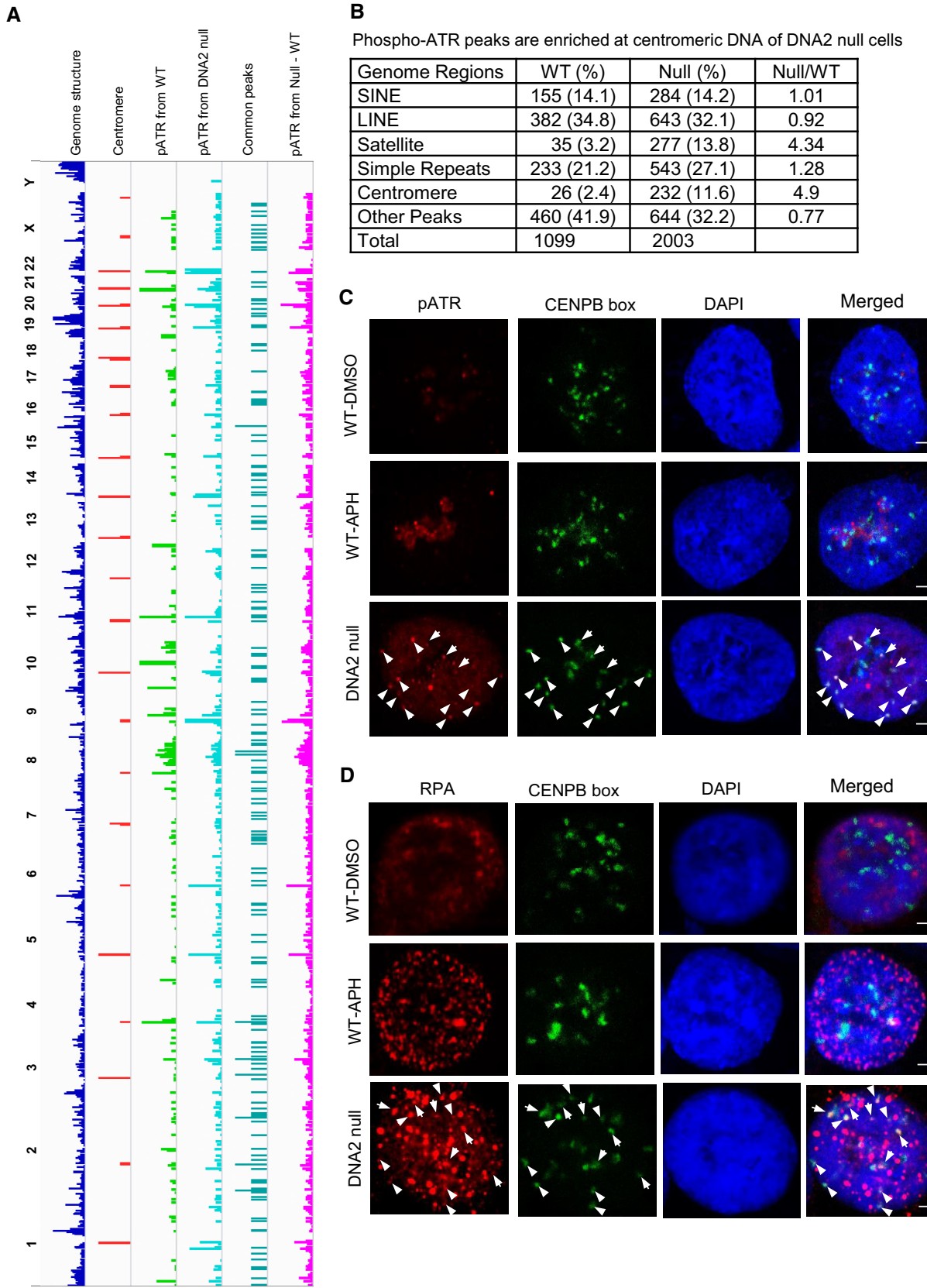

**B** Phospho-ATR peaks are enriched at centromeric DNA of DNA2 null cells

| Genome Regions | WT (%) | Null (%) | Null/WT |
|---|---|---|---|
| SINE | 155 (14.1) | 284 (14.2) | 1.01 |
| LINE | 382 (34.8) | 643 (32.1) | 0.92 |
| Satellite | 35 (3.2) | 277 (13.8) | 4.34 |
| Simple Repeats | 233 (21.2) | 543 (27.1) | 1.28 |
| Centromere | 26 (2.4) | 232 (11.6) | 4.9 |
| Other Peaks | 460 (41.9) | 644 (32.2) | 0.77 |
| Total | 1099 | 2003 | |

**Figure 6.**

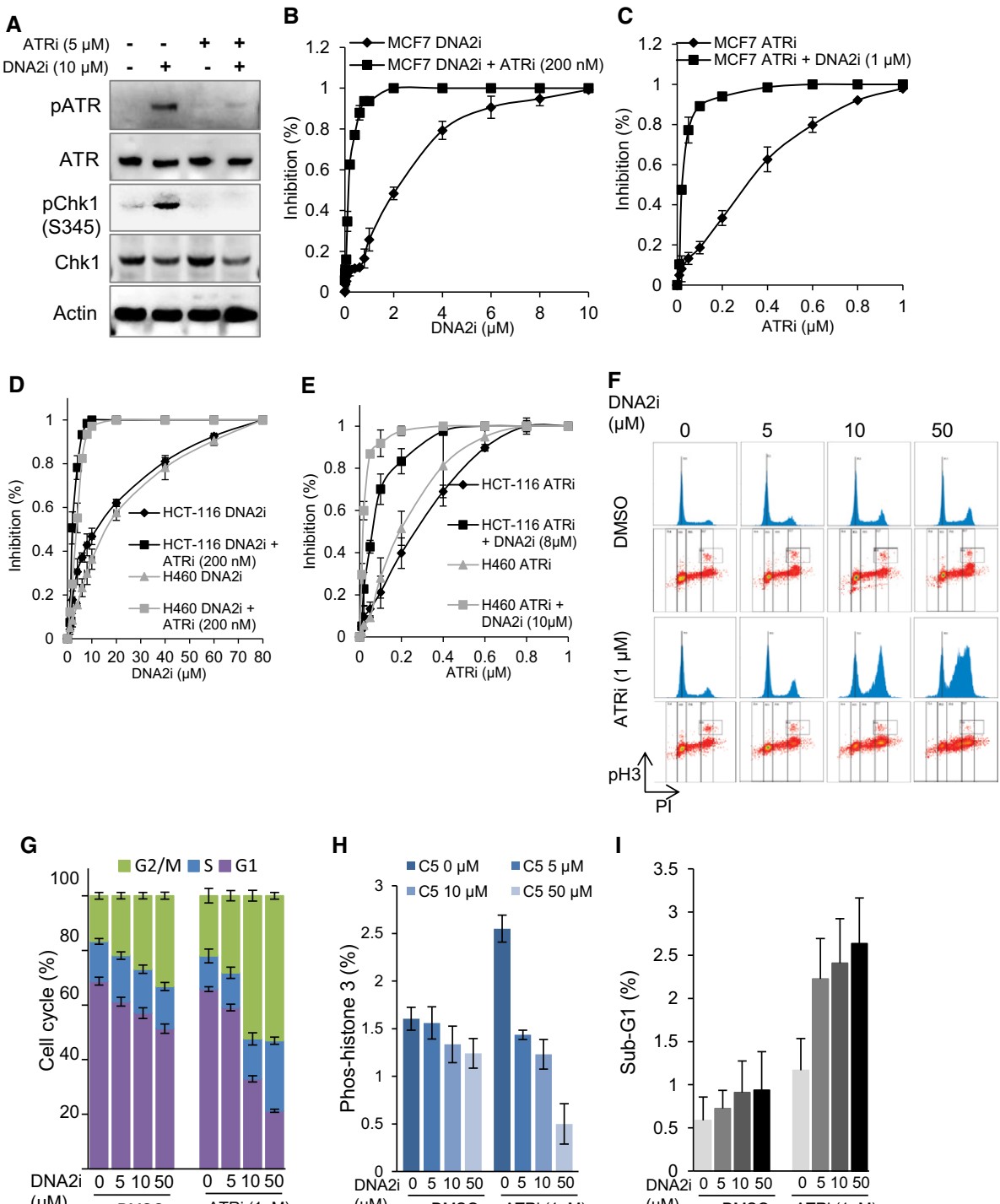

**Figure 7. Synergistic effect on cell death induced by the DNA2 inhibitor C5 and the ATR inhibitor VE-821.**

A    MCF7 cells were incubated with the DNA2 inhibitor C5 (10 μM; DNA2i) for 24 h, and then co-incubated with the ATR inhibitor VE-821 (5 μM; ATRi) for another 24 h, or DNA2i (48 h) or ATRi (24 h) alone. Western blot analysis showing activation of ATR by C5 and inhibition of ATR kinase by VE-821.

B–E    MCF7 cells (B, C) and HCT-116 and H460 cells (D, E) were treated with the indicated drugs as single agents and in combination for 2 weeks. Cell survival was then assessed by colony formation assay. Means ± SD of colonies from three biological repeats are shown.

F–I    MCF7 cells were treated with the indicated DNA2 inhibitors (DNA2i) for 24 h, and then with vehicle control (DMSO) or ATR inhibitor (1 μM, ATRi) for another 24 h. Cells were harvested and stained with phospho-histone H3 and PI to assess mitotic entry (H) and cell cycle distribution (G). The gating of phospho-histone H3 and each phase of the cell cycle is indicated in a dot plot in panel (F). (I) Quantification of sub-G1 cells (apoptotic cells) in MCF7 cells. Values are means ± SD of three independent assays.

Source data are available online for this figure.

DNA (Fig 6). Thus, DNA2 depletion hypersensitizes cells to ATR inhibition (Fig 7). We further demonstrated that both the nuclease and the helicase activities of DNA2 are critical for the completion of DNA replication through the centromeric regions (Fig 4). *In vitro* biochemical analysis revealed that the concerted action of the nuclease and helicase activities allows DNA2 to efficiently remove hairpins and stem loops at the replication fork (Fig 2). Because these highly stable secondary structures are commonly found at centromeric regions, we propose that the DNA2 helicase/nuclease is a specialized facilitator that removes the replication obstacles that arise from repetitive sequences such as those found in the centromeres and telomeres of mammalian cells (Lin *et al*, 2013).

DNA2 and FEN1 are similar in the sense that both are structure-specific nucleases that are involved in Okazaki fragment maturation (Bae *et al*, 2001). However, although this study shows that DNA2 predominantly binds to the centromeric α-satellite DNA regions, FEN1 does not have such a preference. We therefore propose a labor division between FEN1 and DNA2, in which FEN1 primarily functions at non-repetitive sequences while DNA2 functions at repetitive sequences such as those found in the centromeres and the telomeres of mammalian cells (Lin *et al*, 2013). This difference may be because DNA2 has both nuclease and helicase activities, whereas FEN1 only possesses nuclease activity. Although introduction of a point mutation that eliminates the helicase activity of DNA2 kills human cells, the biological function of this helicase activity was previously unclear (Duxin *et al*, 2012; Pinto *et al*, 2016). Our current work suggests that the helicase activity is a prerequisite for nuclease activity. Only when the helicase displaces the DNA fragment to release the ssDNA regions, can the nuclease then act to remove the DNA structures acting as replication obstacles.

Centromeres, as an isolated replication domain, pose a challenge because of their unique repetitive DNA sequences and specialized secondary structures for establishing kinetochores (Aze *et al*, 2016). However, how centromeres are replicated and processed in order to form these specialized structures remains largely unclear. Our current study demonstrates that DNA2 is a novel factor important for centromere replication and CENP-A deposition. We considered that DNA2 may have three distinct roles during centromere replication and processing. First, similar to what we previously observed, DNA2 may use its helicase and nuclease activities to resolve secondary structures such as G4 quadruplex and/or stem loop on DNA templates to promote efficient progression of replication forks. This is supported by evidence that DNA2 knockout causes spontaneously stalled replication forks primarily at centromeres but not non-centromeric regions (Fig 3). Second, DNA2 may work together with FEN1 to remove secondary structures at the single-stranded RNA–DNA primer flaps, which will cause unwanted mitotic recombination if not efficiently removed (Tishkoff *et al*, 1997). Third, DNA2 may play an important novel function in processing centromeric DNA to form specialized structures for loading of CENP-A and inhibiting ATR activation at centromeres. This idea is supported by evidence that maintenance of the intrinsic centromere topological structure by CENP-A and SMC2-4 prevents extensive accumulation of ssDNA under unchallenged conditions and that centromeres accumulate ssDNA in response to DNA replication fork stalling when centromeric DNA organization is disrupted (Aze *et al*, 2016). The ssDNA at stalled centromeric DNA forks may be generated by helicase-driven dsDNA unwinding or nuclease-mediated DNA resection. Interestingly, the helicase and nuclease activities of DNA2 have been shown to be involved in formation of ssDNA during DNA double-strand break repair (Cejka *et al*, 2010; Nimonkar *et al*, 2011). Further studies will clarify the role of DNA2 in formation of ssDNA at centromeres and whether an extended ssDNA region is important for formation of centromere specialized structures for loading of CENP-A and inhibiting ATR.

The late-S/G2 cell cycle arrest and DNA damage checkpoint activation in DNA2-null cells arise, at least in part, due to the incomplete replication of centromeric DNA. Our CHIP-seq and IF-FISH studies both support the idea that centromeric DNA is a dominant region within the genome to which activated phospho-ATR is recruited when cells lack DNA2. Importantly, this is in contrast to replication of centromeric DNA in unstressed WT cells, in which ATR is hypo-activated (Aze *et al*, 2016) in order to facilitate the replication of centromeric DNA. If the activation of ATR in DNA2 null cells is to overcome endogenous replicative stresses at the centromeric DNA region as previously suggested (Buisson *et al*, 2015), targeting both DNA2 and ATR will generate synergy in cancer treatment. Indeed, we observed such synergy in several cancer cell lines (Fig 7). As compared to many chemotherapeutic drugs, which target DNA replication and repair in general, DNA2 inhibitors specifically affect a subset of the genome replication landscape to generate endogenous DNA replication stresses and cause late S-phase arrest. Therefore, DNA2 inhibitors and their potential synergy with other chemotherapeutics may provide new, more effective cancer treatment regimens.

## Material and Methods

### Cell culture, reagents, and antibodies

HCT-116 DNA2$^{Flox/+/+}$ (WT) and DNA2$^{Flox/-/-}$ (inducible deletion) cells were gifts from Dr. Eric Hendrickson at the University of Minnesota. Exon 2 of one copy of the DNA2 gene was flanked by LoxP sites, while the other two copies were either disrupted (DNA2$^{Flox/-/-}$) or intact (DNA2$^{Flox/+/+}$). Cre recombinase was inducible by 1 μM 4-hydroxytamoxifen (4-OHT), which led to excision of DNA2 exon 2. All cells were cultured in Dulbecco's modified Eagle's medium (DMEM) supplemented with 10% FBS and 1% penicillin/streptomycin. 4-OHT (cat# T176) was from Sigma-Aldrich. The DNA2 inhibitor C5 was developed in our laboratory (Liu *et al*, 2016). The ATR inhibitor (VE-821, cat# A2521) was from ApexBio Technology. Antibodies against phospho-Chk1 (S345; cat# 2348), Chk1 (cat# 2360), phospho-histone H3 (S10; cat# 9701), and GAPDH (cat#2118) were from Cell Signaling Technology. The antibody against ATR (cat# sc-1887) was from Santa Cruz Biotechnology. The DNA2 (cat# ab96488) antibody was from Abcam. The CENP-A (cat# GTX13939) and phospho-ATR (T1989; cat# GTX128145) antibodies were from GeneTex.

### Western blot analysis

For Western blot analysis of total cell lysates, cells were incubated with lysis buffer [50 mM Tris pH 8.0, 140 mM NaCl, 1% Triton X-100, 0.05% SDS, 1 mM EDTA, and 1× protease and phosphatase

inhibitors (Roche)] on ice for 10 min, and then centrifuged at $20,000 \times g$ for 10 min at 4°C to clear the lysates. The resulting whole-cell lysates were boiled with 2× SDS loading buffer for 10 min before loading for SDS–PAGE. The antibodies used for Western blot analysis are specified above.

### Protein purification and *in vitro* assays

Purification of DNA2 from 293T cells was done as previously described (Lin *et al*, 2013). For *in vitro* assays, the purified WT and mutated DNA2 proteins were incubated with 1 pmol of 5′- or 3′-labeled DNA substrates in 10 μl reaction buffer, containing 50 mM HEPES-KOH (pH 7.5), 45 mM KCl, 5 mM MgCl$_2$, 1 mM DTT, 0.1 mM EDTA, 2 mM ATP, 200 units of creatine phosphokinase, 0.5 mM NAD, and 5 mM phosphocreatine. The denatured oligonucleotides were then resolved on a 15% sequencing gel and exposed to X-ray films for analysis.

### Immunofluorescence

For immunofluorescence detection of phospho-histone H3 (S10; cat# 9701) and CENP-A (cat# GTX13939), cells were grown on coverslips before the initiation of experimental treatments. After treatment, cells were fixed with 4% paraformaldehyde, permeabilized with 0.25% Triton X-100 in PBS, and blocked with 5% BSA for 1 h at room temperature (RT). Phosphorylated proteins were detected with anti-phospho-histone H3 (S10) or anti-phospho-ATR (T1989), and appropriate fluorescence-conjugated secondary antibodies (Thermo Fisher Scientific). The cells on coverslips were mounted with ProLong Gold anti-fade reagent containing DAPI (Thermo Fisher Scientific) before microscopy.

### IF-FISH

IF-FISH analysis of phospho-ATR, RPA, and CENP-B box was done as previously described (Lin *et al*, 2013). Briefly, cells on coverslips were washed with 0.25% Triton X-100 in PBS on ice for 2 min, fixed with 4% paraformaldehyde, permeabilized, and stained with anti-phospho-ATR (cat# GTX128145) and RPA (cat# ab2175), and appropriate secondary antibodies. Cells were then fixed again using 4% paraformaldehyde for 10 min to preserve the immunofluorescence signal. Then, a standard centromere FISH probe from PNA BIO INC (cat# F3004) was used to hybridize with the DNA and mark the centromere. More specifically, cells were hybridized with 0.3 mg/ml PNA probes (100 nM) targeted to the centromere, in 70% deionized formamide, 10 mM Tris (pH 7.2), and 5 mg/ml blocking agent from Sigma (cat# 11096176001) that was boiled to dissolve. DNA was denatured on a heat block for 10 min at 80°C and incubated at 37°C for 4 h in the dark in a moist chamber before washes. More than 100 cells were analyzed by fluorescence microscopy to assess the co-localization of phospho-ATR and centromeres.

### siRNA

DNA2 siRNA duplexes were purchased from Life Technologies (cat#31053582). The siRNA transfection reagent was purchased from Polyplus Transfection. siRNA-mediated knockdown of DNA2 was done following the manufacturer's instructions. Briefly, cells were grown to 30–40% confluence and washed with FBS- and antibiotic-free medium. siRNA duplexes were mixed with FBS- and antibiotic-free medium and incubated with transfection reagent for 10 min. This siRNA/reagent mixture then was added to cells in FBS- and antibiotic-free medium. After 5–7 h of incubation, cells were incubated for 72 h in a final concentration of 10% FBS and 1% penicillin/streptomycin added into the transfection medium before cell harvesting.

### DNA2 complementation

In our hands, overexpression of WT DNA2 using the p3xFLAG-CMV DNA2wt (Lin *et al*, 2013) and pBabe hygro 3xFLAG DNA2wt vector (Duxin *et al*, 2012) kills host cells. Therefore, to complement the DNA2 knockout, we cloned the DNA2 cDNA into the pLenti PGK Hygro DEST (Addgene cat# 19066) vector, in which DNA2 expression is under control of the weaker PGK promoter. Specifically, WT DNA2, as well as the D294A ND and K671E HD DNA2 mutants, was PCR amplified from our original p3xFLAG-CMV vector using the following primers: DNA2 pLenti-F TTCCGGCTGCGTCCAGGATGGAGCAG and DNA2 pLenti-R CTGGCTGCCTTATTCTCTTTGAAAGTCACCCAATA TGTGG. After removal of the primers with the Qiagen PCR purification kit (cat# 28104), the cDNAs were in-fusion (Clontech)-cloned into a linearized pLenti PGK Hygro DEST vector that was generated by PCR using the primers pLenti-ATTR2-F GAATAAGGCAGCCAGTCTGC AGGTCGA and pLenti-ATTR1-R TGGACGCAGCCGGAAGCATAAA GTGTAAAGC. After transfection with the pLenti PGK Hygro DEST-DNA2, the DNA2$^{Flox/-/-}$ host cells survived hygromycin B selection for the complementation experiments shown in Figs 3 and 4.

### Metaphase cell immunofluorescence staining

To perform immunofluorescence staining on metaphase cells, cover slips were coated overnight in 50 μg/ml collagen (Sigma cat# C5533). The next day, the collagen solution was aspirated and the slides dried. 4-OHT (1 μM) was added to the medium during the following entire cell culture procedure and co-incubated with drugs (5 mM thymidine and 10 μM RO-3306) that were later added to synchronize the cells. The cells were then seeded at approximately 50–70% confluence, and 5 mM thymidine was added to the media during seeding. After 24 h, the thymidine was washed away using fresh medium, and the cells were released into fresh media with 10 μM RO-3306 (Sigma cat# SML0569). After another culture period of 24 h, the cells were released into fresh warm medium for 45–60 min before fixation with 100% methanol. After fixation, the slides were blocked (5% FBS, 0.3% Triton X-100 in PBS) for at least 3 h at RT, immunostained with the indicated primary antibodies, and incubated with Cy5-, Alexa Fluor 488-, and FITC-conjugated secondary antibodies. The immunostained slides were mounted in Vectashield mounting medium containing DAPI.

### Chromosomal mis-segregation analysis

Chromosomal mis-segregation was assessed using microscopy analysis of cells on coverslips. Briefly, nocodazole (100 ng/ml) was added for 3 h to enrich for metaphase cells. One hour after release from

nocodazole, the cells were fixed, stained with DAPI and anti-CENP-A, and mounted with gold anti-fade reagent (Invitrogen cat# P36930). The percentages of cells with segregation abnormalities were quantified and statistically analyzed (> 100 cells for each group).

### Flow cytometry analysis of cell cycle and mitotic entry

For analysis of cell cycle, cells were fixed in 70% cold ethanol for a minimum of 1 h at −20°C. The fixed cells were then centrifuged at $10,000 \times g$ for 10 s. Pellets were resuspended in propidium iodide (PI) solution (PBS with 10 μg/ml PI and 100 μg/ml RNase; Thermo Fisher Scientific) and incubated for 30 min at 37°C. Thirty thousand events were analyzed using a Beckman Coulter CyAn flow cytometer to measure DNA content. The cell cycle distributions were determined using Summit 5.4 software.

For PI and phospho-H3 double staining, approximately $1 \times 10^6$ cells were trypsin-harvested. Cells were then fixed with 70% ethanol at −20°C for at least 1 h. For permeabilization and blocking, cells were suspended in 1 ml of PBS containing 0.25% Triton X-100 and 2% BSA and incubated on ice for 20 min. Cells were then centrifuged at $600 \times g$ for 5 min. The pelleted cells were resuspended in 200 μl of TBS/2% BSA containing anti-phospho-H3 antibody (s10; 1:200 dilution) and incubated for 1 h at room temperature. Cells were then washed with TBST buffer, centrifuged, and stained in TBST/2% BSA containing goat-anti-rabbit IgG FITC (Thermo Fisher Scientific Inc, 1:100 dilution) for 30 min at room temperature in the dark. The cells were washed three times with TBST (1 ml) and stained with PI [5 μg/ml in 300 μl PBS with RNaseA (100 μg/ml)] for 30 min at 37°C in the dark. Cell cycle phase and phospho-H3 were analyzed using a CyAn flow cytometer. Forty thousand gated events were collected, and the percentage of mitotic cells was determined using Summit 5.4 software.

### Nuclear histone extraction

Cells were harvested, washed twice with ice-cold PBS, and then resuspended in Triton extraction buffer (TEB; PBS containing 0.5% Triton X-100 [v/v], 1× protease and phosphatase inhibitors, 0.02% [v/v] NaN₃) at a density of $10^7$ cells/ml. The cells were lysed on ice for 10 min with gentle stirring and centrifuged at $500 \times g$ for 10 min at 4°C. The cell pellets were washed in 500 μl of TEB, resuspended in 0.2 N HCl at a cell density of $4 \times 10^7$ cells/ml, and incubated overnight at 4°C. The remaining chromatin was cleaned at $20,000 \times g$ for 10 min at 4°C. The protein content of the supernatants was determined before Western blot analysis.

### Chromatin immunoprecipitation (ChIP)

Cells grown on 10-cm dishes were cross-linked with 1% paraformaldehyde for 5 min, shaken in a final concentration of 125 mM glycine for 5 min, washed twice with PBS, and collected by scraping. After centrifugation at $600 \times g$ for 5 min, the cell pellets were resuspended at $10^7$ cells/300 μl in 0.5% SDS IP buffer [50 mM HEPES pH 7.5, 150 mM NaCl, 1 mM EDTA, 0.5% SDS, 1.0% Triton X-100, 5 mM NaF, and cOmplete protease inhibitor (Roche)] on ice for 10 min and then sonicated to break the DNA into 150- to 300-bp fragments. After clearance by centrifugation at $20,000 \times g$ for 10 min, 100 μl of lysate was either saved as input or was mixed with 900 μl of 0.1% SDS IP buffer [50 mM HEPES pH 7.5, 150 mM NaCl, 1 mM EDTA, 0.1% SDS, 1.0% Triton X-100, 5 mM NaF, and cOmplete protease inhibitor (Roche)]. Then 1 μg of DNA2 (Abgent, cat# AP10182c) or Fen1 (GeneTex, cat# GTX70185) antibodies and 25 μl of pre-prepared Dynabeads (Thermo Fisher Scientific) were added to each 1,000 μl of lysate mixture and incubated overnight at 4°C. The beads were captured using a magnetic rack, sequentially washed, and resuspended in 400 μl of Q-elution buffer (20 mM Tris–HCl pH 7.5, 150 mM NaCl, 5 mM EDTA, 1% SDS, and 25 μg proteinase K) overnight at 65°C to elute the DNA. To prepare the genomic input DNA, 100 μl of clean lysate was mixed with 300 μl of Q-elution buffer to elute the DNA, as was done for the ChIP samples. Then, the DNA in the mixed samples was extracted with phenol/chloroform/isoamyl alcohol (pH 7.8), precipitated, washed with 70% ethanol, and resuspended in nuclease-free water for high-throughput DNA sequencing. Immunoprecipitation of the target proteins was verified by Western blot analysis before subsequent whole-genome sequencing or real-time qPCR analysis (Fig EV1).

### Immunocapture and isolation of BrdU-labeled DNA

For the BrdU incorporation experiments, HCT-116 DNA2$^{Flox/−/−}$ cells were treated with 4-OHT for 24 h, a time point for which there was no significant accumulation of cells in late S-phase. Cells were then incubated in 10 μM BrdU for 32 h, which is sufficient time for a full normal cell cycle and BrdU should be incorporated into every cell. Fifty-six hours after 4-OHT treatment, cells had hypothetically finished replication and arrested in late S-phase and were harvested. Cells were then washed with PBS and cross-linked with 1% formaldehyde (Thermo Scientific cat# 28908) in PBS for 5 min on a shaker at RT. After 5 min, glycine was added at a final concentration of 125 mM and cells were incubated for 5 min with shaking. The cells were then washed 2× with PBS, scraped, and collected by centrifugation. To generate 150- to 300-bp DNA fragments, cells were incubated with 0.5% SDS IP buffer [50 mM HEPES pH 7.5, 150 mM NaCl, 1 mM EDTA, 0.5% SDS, 1.0% Triton X-100, 5 mM NaF, and cOmplete protease inhibitor mix and Pefabloc (Roche)] on ice for 10 min. The cell solutions were then transferred into 1.5-ml Bioruptor Pico microtubes and sonicated using 10 cycles of 30-s on and 30-s off in a Bioruptor Pico sonication device. For DNA extraction, 300 μl of Q-elution buffer was added to 100 μl of the sonicated samples, and the samples were further digested with proteinase K at 65°C overnight. DNA was extracted with phenol/chloroform/ isoamyl alcohol (Sigma-Aldrich cat# P2069) for further purification by centrifugation.

Separation of BrdU-labeled DNA from unlabeled DNA was done as previously described (Roux-Michollet *et al*, 2010; Viggiani *et al*, 2010). Briefly, 20 μl of DNA at ~ 20 ng/μl in PBS was heat denatured at 100°C for 1 min and transferred to an ice–ethanol bath for 30–50 s to flash freeze. Tubes of DNA were then thawed completely at RT (< 3 min), mixed with 2 μl of a BrdU antibody (BD Biosciences, cat# 347580), and incubated at RT for 30–40 min in the dark with occasional mixing. The BrdU-labeled DNA, bound with antibody, was then isolated with 20 μl of prepared Dynabeads Sheep anti-Mouse IgG (Thermo Scientific cat# 11031). The BrdU-negative DNA was analyzed by sequencing.

## Single-molecule analysis of replicated centromeric DNA

### DNA combing

All reagents were purchased from Genomic Vision, and DNA combing was performed using their instructions. Briefly, DNA2$^{Flox/+/+}$ and DNA2$^{Flox/−/−}$ cells were treated with 1 μM 4-OHT for 48 h to remove DNA2. Cells were then labeled with 40 μM IdU for 4 h followed by 200 μM CldU for 4 h. DNA was extracted from agarose plugs, washed, and combed onto coverslips. Coverslips with combed DNA were baked at 60°C for 2 h before hybridization.

### Hybridization and DNA track detection

DNA on coverslips was denatured in 0.5 M NaOH + 1 M NaCl solution for 8 min at RT, washed three times with PBS, and dehydrated in 70, 90, and 100% cold ethanol for 1 min each. After drying, DNA on coverslips was hybridized with Vysis CEP 10 SpectrumAqua centromeric DNA probes (Cat# 06J54-020) in the provided buffer to target the centromeric DNA of chromosome 10. After an overnight hybridization at 37°C, coverslips were washed three times in 50% formamide, 10 mM Tris (pH 7.2), and 0.1% BSA (10 min each) and subsequently washed three times with TBST (0.1 M Tris pH 7.2, 0.15 M NaCl, and 0.08% Tween-20). To detect replicated DNA tracks, coverslips were blocked with 5% BSA-TBST for 1 h at RT, incubated with appropriate anti-BrdU primary antibodies (Abcam, clone BU1/75 [ICR1] for detection of CldU and BD Biosciences, clone B44 for detection of IdU) for 1 h at 37°C and then Alexa Fluor-conjugated secondary antibodies (Thermo Fisher Scientific) for 1 h at RT. Coverslips were mounted with prolong gold anti-fade reagent (Invitrogen Cat#P36930) before visualization. Micrographs were taken of all centromeric DNA fibers (including surrounding fibers) on coverslips to assess the progression of DNA replication fork in centromeric versus non-centromeric DNA, using a Zeiss AxioCam 506 Mono.

## Real-time quantitative PCR

The relative centromeric and non-centromeric DNA levels from DNA2 and FEN1 ChIP analyses (Fig 1B) or from genomic DNA (Fig 4A) were determined by qPCR using Power SYBR Green PCR Master Mix (Thermo Fisher Scientific) on a CFX96 Real-Time System (Bio-Rad). Conditions included one cycle of 95°C 10 min followed by 40 cycles of 95°C for 20 s, 60°C for 60 s, and generation of melting curves. The primers used for qPCR are listed in Table EV1.

## High-throughput sequencing data analysis

The Hg38 genome, genome structure, and centromere locations were downloaded from the UCSC genome browser database. The sequencing results were aligned to the Hg38 genome using Novoalign with default settings, only allowing unique alignment. Sequencing data processing and peak calling were done using custom R scripts and the Bioconductor package "chipseq". The aligned reads were extended to 200 bp and subjected to peak calling to identify enriched regions using the corresponding control sample (IgG for DNA2 and genomic input for BrdU) as baseline. The peak identification criteria were as follows: (i) fold change between enriched and control samples > 2.5-fold; (ii) at least 50 reads in the peak; and (iii) minimum peak width ≥ 100. The overlapping peak regions

among the two enrichment methods were identified, and those with > 100 bp width were considered common peaks. The genomic sequences of these common peaks were retrieved, and over-represented motifs were identified using the "peak-motifs" module at RSAT online tools (http://rsat.sb-roscoff.fr/; Medina-Rivera *et al*, 2015), with word sizes of 6, 7, and 8. The top 10 motifs were reported, along with their frequency in the peaks (Fig EV2).

The relative enrichment of centromere peaks was evaluated by comparing two fractions: One was dividing the number of bases in the peaks within the centromere region by the total bases in the centromere regions, and the other is dividing the peaks not in the centromere region by the total bases in the genome that are not in the centromere region. This ratio represented whether peaks were enriched in the centromere region. The *P*-value was calculated using Fisher's exact test.

## Colony formation assay

Briefly, 500 cells were seeded in a 6-well plate and incubated in culture medium containing DMSO or indicated concentrations of drugs. Culture medium, with or without drugs, was changed every 3–4 days. For clonogenic analysis, plates were washed with PBS after 14 days of culture, colonies were fixed and stained with crystal violet solution, and the number of colonies (> 50 cells) counted. The combination index, indicating the synergistic effect of the compounds, was measured using a previously published method (Chou, 2010).

## Statistics and reproducibility

All data used for statistical analysis were from at least three independent biological repeats. Unpaired two-tailed *t*-tests were performed to compare differences between two groups. One-way ANOVA was used to determine differences among the means of three or more groups. An unpaired two-tailed *t*-test was performed to confirm where the differences occurred within groups. A *P*-value < 0.05 was considered significant and is indicated by "*" in figures.

## Data accessibility

The whole-genome sequencing data from this publication for generating Figs 1A and 6A have been deposited to the NCBI Gene Expression Omnibus database https://www.ncbi.nlm.nih.gov/geo/ and assigned the accession number GSE108619.

**Expanded View** for this article is available online.

## Acknowledgements

This project was supported by R01CA085344 to B.H.S and R50CA211397 to L.Z. We would like to thank Dr. Eric Hendrickson for providing the HCT-116 DNA2 cell lines and Dr. Judith Campbell for productive discussions during early stages of the project. We are grateful to Dr. Yilin Liu and Dr. Lufen Chang for critical suggestions on the experimental design. We thank Jocelyn Rodrigues, Huifang Dai, Wenpeng Liu, and Ayesha Ng for technical assistance. We also thank the staff of the Integrative Genomics Core for high-throughput DNA sequencing and data analysis and the staff of the Flow Cytometry Core for technical assistance, both of which are supported by the City of Hope Comprehensive Cancer Center Support Grant (P30CA33572). We thank Dr. Keely Walker for editorial assistance.

## Author contributions

ZL, LZ, and BS designed most of the experiments. ZL conducted the majority of the experiments. BL and ZS designed and performed the metaphase cell staining and microscopy. WJ helped with the chromosomal segregation and other experiments. XW conducted high-throughput DNA sequencing and data analysis with the assistance of VWL, MZ, and AG helped genetic manipulation and validation of the cell lines used in the current study. BS, ZL, and LZ contributed to manuscript preparation. BS supervised the entire project, designed and coordinated the experiments, and supervised manuscript preparation.

## Conflict of interest

The authors declare that they have no conflict of interest.

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
