## [Review Process File · The EMBO Journal]

hDNA2 nuclease-helicase promotes centromeric DNA replication and genome stability

Zhengke Li, Bochao Liu, Weiwei Jin, Xiwei Wu, Mian Zhou, Vincent Wenzhe Liu, Ajay Goel, Zhiyuan Shen, Li Zheng and Binghui Shen

Review timeline:

Submission date:	13 February 2017
Editorial Decision:	9 March 2017
Appeal/Resubmission:	4 December 2017
Editorial Decision:	16 January 2018
Revision received:	7 March 2018
Editorial Decision:	10 April 2018
Revision received:	16 April 2018
Accepted:	25 April 2018

Editor: Hartmut Vodermaier

Transaction Report:

1st Editorial Decision

9 March 2017

Thank you for submitting your manuscript on DNA2 roles in centromeric DNA replication for our editorial consideration. It has now been assessed by three expert referees, whose reports are copied below for your information. I am afraid to say that in light of these reports, we have decided that we cannot consider your study further for publication in The EMBO Journal. As you will see, the referees appreciate the interest of the addressed question, as well as your demonstration of DNA2 presence at centromeres and ability to act on centromeric repeat-mimicking DNA substrates. However, they all remain unconvinced that cellular follow-up studies are sufficiently conclusive and offer sufficiently definitive insights into the functional importance of DNA2 at centromeres in cells. I will not go through the referees' individual concerns in detail here, as they are well explained in all three reports; however, I am afraid that the generally shared criticisms and lack of strong support from any of these three trusted experts leave me little choice but to conclude that this work is presently not a strong and promising candidate for an EMBO Journal article.

In any case, thank you again for the opportunity to consider this work for our journal. I am sorry for not being able to come to a more positive conclusion on this occasion, but nevertheless hope that you will find our referees' detailed comments and suggestions helpful, and wish you every success in publishing this work.

REFeree REPORTS

Referee #1:

In this manuscript, the authors uncover a novel function of the human DNA helicase/nuclease in

centromeric DNA replication. In particular, these studies propose DNA2 is specifically required for Okazaki fragment maturation at regions containing repetitive sequences, such as centromeres and telomeres. The authors used chromatin immunoprecipitation approaches coupled to whole-genome DNA sequencing to show that DNA2 is preferentially localized to centromeric alpha-satellite regions. Next, they combined biochemical and cellular approaches to show that both the nuclease and helicase activities of human DNA2 are required to facilitate Okazaki fragment maturation on centromeric DNA. Loss of DNA2 function leads to ATR/ATM checkpoint activation, late-S cell cycle arrest and chromosome segregation defects. Moreover, inhibition of the ATR checkpoint prevents S-phase arrest and forces DNA2 knockout cells to enter mitosis, thereby leading to mitotic catastrophe. Finally, the authors show that the combined inhibition of DNA2 and ATR activity has a synergistic effect on cancer cell survival pointing to a new strategy for cancer cell killing. Collectively, this is an interesting study that proposes a novel function of the DNA2 nuclease/helicase in replication. There are, however, several major concerns that the authors should properly address to strengthen their results and support their conclusions.

Major concerns:

1. The major criticisms of this reviewer is that while the authors provide compelling evidence for a role of DNA2 at centromeric regions, their conclusion that the observed defects in cell cycle progression and cell survival (when combining DNA with ATR inhibition) is mainly due to loss of DNA2 function in centromeric DNA replication is overstated and not supported by the data. As the authors state in their manuscript, DNA2 has additional roles in DNA replication and DSB repair. Thereby, DNA2 knockout will suppress all these additional DNA2 functions. Separation of function DNA2 mutants that suppress DNA2 function at centromeric regions, while sparing its previously described function in replication and repair, would be necessary to support the authors' conclusion. While generating separation of function mutants is probably a difficult task, the authors should at least revisit their conclusions to mention how the previously reported functions of DNA2 might contribute to the observed phenotypes.
2. Figure 3B and 3C. The authors should include immunoblots of total protein levels, in addition to chromatin-bound fractions, to rule out the possibility that DNA2 loss affects CENPA expression.
3. The authors should repeat all their cell cycle analyses by co-staining S-phase cells and DNA content because staining DNA content using PI alone does not allow to distinguish between late S and G2/M phases of the cell cycle. Also, the authors should quantify the Sub-G1 fraction in order to test whether these cells are permanently arrested or have undergone cell death (and therefore go to the Sub-G1 fraction).
4. Figure S2. As mentioned above, the authors should co-stain S-phase cells and DNA content (BrdU-PI) to better distinguish between late S and G2/M phases of the cell cycle. Moreover, Figure S2C does not show any significant difference between the cell cycle profiles of DNA2 WT and DNA2 knockout cells. This cell cycle analysis should be repeated at a time point where DNA2 loss leads to a more marked G2/M phase arrest (for example 56 or 72 hours, as shown in Figure S2A).
5. The authors state that DNA2-null cells, but not WT cells, accumulate in late S/G2 phase. However, Figure S2A shows only the quantification of cells in G2/M without quantifying the percentage of cells in late S/G2.
6. Figure 4C. Why does treatment with RO-3306 lead to a decrease in DNA2 protein levels? The authors should use nocodazole as an alternative approach to accumulate cells in G2/M and test whether they obtain reproducible results.
7. To explain the molecular basis for ATR and ATM checkpoint activation observed upon DNA2 loss, the authors state ATR activation may result from the inability of DNA2 to resolve centromeric DNA replication intermediates whereas "ATM is constantly activated by endogenous double strand breaks resulting from defects in DNA2-mediated end resection". What is the actual evidence for this conclusion? How frequent are DSBs in DNA2-depleted cells? The authors should compare the amount of DSBs in DNA2^{Flox/+} versus DNA2^{Flox/-} cells to support their conclusion.
8. The authors state on page 13 that "a small population of DNA2-null cells spontaneously entered

mitosis". This conclusion is based on the observation that DNA2 loss causes a reduction of phospho-histone 3 positive cells from 1.5% to 0.48%. However, these numbers are very small and the observed differences could be simply associated to small differences in the density of cells in the wells or in the timing of seeding or trypsinization.

9. Figure 6A. The authors should provide the same histograms including the Sub-G1 fractions to rule out the possibility that the observed decrease in late S/G2 phase of the cell cycle is due to increased cell death.

10. Figure 6C. The authors should include better images for the metaphase-spreads. In particular, the metaphase-spreads of the DNA2Flox^{-/-} cells without treatment already show some significant defects compared to control as the chromosomes seem to be more condensed. Is this because of the low picture quality or is a real phenotype? In addition, the authors should use centromere specific probes to investigate whether there are any specific defects at centromeres on the metaphase-spreads.

11. Figure 6C. The authors should include in the same figure the data obtained with DNA2Flox^{-/-} cells without inhibitor treatment (4-OHT + ; ATMi/ATRi -).

Referee #2:

In this manuscript, Li et al. describe the role of DNA2 role on centromeric DNA replication and stability. In general, the manuscript is interesting especially for what concerns the first part in which the authors suggest that DNA2 is binding centromeric DNA, promoting its replication. To support these claims the authors show that that in vitro DNA2 is specifically involved in resolving hairpin structures potentially arising in centromeric DNA. Furthermore, the authors show that DNA2 depletion in cells lead to incomplete centromeric DNA replication, activation of the DNA damage response and mitotic entry failure. Although the first part concerning the involvement of DNA2 in centromere stability is acceptable, the experiments related to consequence of DNA2 deficiency on cell cycle progression are quite disappointing and make little sense considering the role proposed for DNA2 at centromeres. This second part is indeed all based on minimal differences of pH3S10 and blots for phospho Chk1 and Chk2 of poor quality and not well controlled. This unfortunately prevents me to give a positive evaluation of this work.

Critiques

-The experiments presented in the first paragraph only show that DNA2 binds centromeric DNA and do not directly prove that it is required for its replication. Direct evidence for DNA2 involvement in centromeric replication is missing. The Chip-seq experiment should be carried out also on BrdU positive strand to address whether the DNA2 is also present on replicated sequences.

-It is not clear whether the observations related to under-replication of centromeric DNA are due to the absence of DNA2 action on hairpins or to checkpoint activation. ATRi should be used to clarify this issue.

- Binding of DNA2 to centromeric regions is not supported by alternative approaches such as colocalization with centromere markers by immunofluorescence (using for example anti CENP-A antibodies or fish probes for satellite DNA).

-The consensus sequence for the major repeats found at centromere is not novel. The authors did not cite important works, which clearly identified a similar consensus sequence underlying the formation of stem-loop structures (Grady D. et al, PNAS 1992, 89, 1695-1699; Catasti P. et al Biochemistry, 1994, 33, 3819-3830). This somehow diminishes the novelty of these findings.

-Some of the repeats resemble telomeric sequences. Since DNA2 binds also TRF1 (Lin et al., Embo J. 2013), can the authors exclude that the sequence belongs to telomeric regions instead than to centromeres?

-It is not clear how the fold enrichment is calculated in Fig 1c.

-There are a number of inconsistencies in figure 3-4 and related panels.

First, in S2 figure (panel A and B), the authors show that after depletion of DNA2 the intra-S-phase checkpoint is activated and as evidence they show Chk1 phosphorylation levels. However, there is no accumulation in S-phase in the FACS profiles shown.

Second, the authors suggest that Chk1 activation is due to centromeric defects arising in the absence of DNA2. However, this claim makes no sense as DNA2 is known to act in many other genomic regions, being required for the processing of double strand breaks and long flaps of okazaki fragments during DNA replication. Therefore, activation of Chk1 could well be due to lack of functional DNA2 in different parts of the genome.

Third, the effects on CENP-A loading (Fig S3 panel B) could be due to the accumulation of cells in G2/M, rather than under-replication of centromeric regions.

Fourth, the authors should improve the quality of the blots, which are rather poor.

-I disagree with the interpretation of Figure 4B and 4C as they do not provide enough evidence to claim that there is a block in S-phase. Cells seem to proceed slowly in S-phase and then progress into G2. Increasing levels of CyclinB and decreasing levels of CyclinE are consistent with this different interpretation.

-Blots in Fig4C ed S2B are only done with DNA2^{-/-} cells. They should be performed with DNA WT cells, which also show high signal for phosphoChk1 (Fig 6B)

-In the figure 4A, the authors should quantify the mitotic entry capacity of DNA2 null cells by FACS analysis instead of Immunofluorescence

-Immunofluorescence images are of extremely poor quality, which does not seem to be due to file compression. Better quality images should be provided.

-In Fig 5C the authors calculate a congressional index. This parameter is not informative. They should instead calculate the percentage of cells that have all the chromosomes aligned on the metaphase plate as it is usually done. They should also specify that they are monitoring cells after nocodazole release (this is only mentioned in materials and methods). It would be worth adding MG132 to block anaphase before calculating the percentage of cells with aligned chromosomes as congression delay could be simply due to poor recovery from nocodazole mediated arrest.

Figure 6:

Panel A is confusing. The blots in panel B are clearly run on different gels. This should be specified. Are all the samples from DNA2 null cells? Looking at the WB of DNA2 it is not possible to assess the presence-absence of DNA2. In panel C a graph should be added. Also the authors should clarify how they measured mitotic catastrophe.

Figure 7:

Panel A shows the inhibition of DNA2 in Breast cancer cells. Could the authors justify the lack of any cell cycle block in the presence of active Chk1 and Chk2 proteins after the addition of DNA2 inhibitor (Panel B)? Also, these experiments should be repeated using a panel of cancer cell lines instead of one cell line. DMSO treated cells should be included as control.

Again, the blots shown in panel B seem to be run on different gels as separate experiments.

Referee #3:

Li and colleagues outline work in this manuscript that the authors conclude shows that hDNA2 is responsible for centromeric DNA replication. They base their conclusion on cellular sequencing results that show high occupancy of DNA2 at centromeres. This observation is followed up by mechanistic studies wherein the authors used random and centromeric DNA substrate mimics to assess the activity of Dna2 on these substrates. Additional cellular results show that the ATM/ATR pathway is activated due to incomplete centromeric replication. The fundamental conclusion of the

study is that since DNA2 is important for centromeric replication, it can be used as a chemotherapeutic target along with ATR inhibitors in the treatment of cancer.

There are considerable issues with the interpretation of results in this study as presented. Primarily, centromeric DNA requires the help of MMR proteins to help resolve the looped structures. Hence choice of the HCT116 cell line (which are deficient in MLH) is an odd choice for these studies. Secondly, there is no information about how the sequencing libraries were made in the materials and methods or how many times replicates were done in the sequencing analysis? What was the read size, and was this paired-end sequencing? Was the chromatin fragmented by sonication or micrococcal nuclease digestion? Also for the sequencing data that is presented in the study, was it merged data from multiple samples or from a single experiment? How reproducible were the results? Data from individual sequencing needs to be added in the supplementary results to show reproducibility. The data with DNA2 flox^{-/-} cells and response of the repair pathways is not completely novel, with even the authors acknowledging their data is consistent with previous findings. The results presented in this manuscript remain purely observational without taking into consideration the confounding effects of the role of DNA2 replication of the whole genome.

Major Concerns

Figure 1: While it is hard to judge the sequencing data without having any information on the exact methodology used to generate the data or its reproducibility, there are other considerations for the observation that need to be either analyzed or discussed. For example, the conclusion from the sequencing data that DNA2 was involved in centromeric replication is solely based on higher occupancy of DNA2 compared to FEN1. While FEN1 cannot resolve fold-back intermediates it can do so in the presence of other helicases, especially the 5'-3' RNA/DNA helicase, Pif1. Were other proteins involved in the Okazaki fragment maturation considered in the study, for example, Pif1, WRN and RPA?

Fig S1: How efficient was the depletion experiment? Are the authors confident about not having any contaminations in the depleted pool of cells?

Figure 2: Templates having the 5' fold required ~15-20 fold excess of Dna2 to resolve the structure. While this was mentioned in the materials and methods section, it should be included in the actual results, so that the reader can immediately grasp the requirement for higher concentration of protein for fold back flap resolution. Interestingly, the wild type protein did not show any cleavage below the 5' terminal product, suggesting the helicase activity of Dna2 was not sufficiently strong enough to create a ligatable nick. Does RPA aid in this function of Dna2 by melting out some of the structure. Based on data from Aze et al, 2016, RPA abundance was lower at centromeric DNA hence assessment of abundance of RPA at the centromere in this system would have been helpful for the in vitro study.

The cleavage products should contain quantitation. It is intriguing that the 5' and 3' labeled bubbled centromeric substrates were resolved with different efficiencies by the wild-type enzyme, though it should have been the same.

Did the authors try test ligation of the resolved fold back flap substrate to ensure DNA2 could process these structures in the absence of FEN1?

Figure 3: What is the consequence of overexpression of FEN1 in the DNA2-null cells?

Figure 4-5: Arrest of DNA2 cells in the S or G2 phase and activation of the ATR/ATM pathways could be the consequence of its role in nuclear Okazaki fragment maturation and may not necessarily only be caused due to delays in centromeric DNA replication

Thank you for your kind response to my request to re-evaluate our amended work by EMBO Journal. It takes longer than I anticipated to compile the point-by-point response letter. I am sorry that I am only able to get it back to you today. Over all, for the current revision, we have developed the Single Molecular Analysis of Replicated DNA (SMARD) assay to directly assess the role of DNA2 in the centromeric DNA replication (Figure 3-new). This is the first time in the field a technology has been developed and applied to examine centromeric DNA replication statuses. It has offered insights into the functional importance of DNA2 at centromeres in cells and helped us to address many of the concerns that the reviewers have raised. In addition, our phosphor-ATR ChIP-seq and IF-FISH (Figure 6-new) clearly show that the centromere is the major region where replication-stalling signals occur when DNA2 is knocked out. We have done/redone a vast majority of the experiments that the reviewers have suggested including the DNA2 centromere functions in MMR-proficient or deficient cells, western blotting with high quality, and synergistic effects of

inhibitors in multiple cell lines. However, we did attempt to do the suggested experiments to distinguish the specific cell cycle phases that DNA2 null mutant cells were arrested in order to focus our efforts on our major claims. Coupled with an unbiased whole genome screening strategy including DNA2 ChIP-seq/BrdU incorporation negative DNA sequencing, we were able to systematically study the role of DNA2 nuclease-helicase in centromere replication in the current work. In the response letter, we have explained all of these changes in response to the reviewers' comments. We wish you will give us another chance for the reviewers to re-evaluate our revised manuscript. Besides the response letter, I have also attached the updated manuscript PDF file containing the text, figures and supplemental materials as a single file for your convenience.

I would like to sincerely thank you for your time and efforts for re-considering our amended manuscript.

Point-by-point responses to the reviewers' comments:

We would first like to thank the expert reviewers for their comprehensive reviews on our initial submission and for giving constructive suggestions for us to improve the quality of the data and manuscript. For the last 9 months, we have developed the Single Molecular Analysis of Replicated DNA (SMARD) assay to directly assess the role of DNA2 in the centromeric DNA replication (Figure 3). This is the first time in the field a technology has been developed and applied to examine centromeric DNA replication statuses. It has offered insights into the functional importance of DNA2 at centromeres in cells and helped us to address many of the concerns that the reviewers have raised. In addition, our phosphor-ATR ChIP-seq and IF-FISH (Figure 6) clearly show that the centromere is the major region where replication-stalling signals occur when DNA2 is knocked out. We have done/redone a vast majority of the experiments that the reviewers have suggested including the DNA2 centromere functions in MMR-proficient or deficient cells, western blotting with high quality, and synergistic effects of inhibitors in multiple cell lines. Coupled with an unbiased whole genome screening strategy including DNA2 ChIP-seq/BrdU incorporation negative DNA sequencing, we were able to systematically study the role of DNA2 nuclease-helicase in centromere replication in the current work. Based on all of these important amendments, we wish that the reviewers will re-evaluate and consider our revised manuscript. The following are point-by-point responses to the reviewers' comments.

Referee #1:

In this manuscript, the authors uncover a novel function of the human DNA helicase/nuclease in centromeric DNA replication. In particular, these studies propose DNA2 is specifically required for Okazaki fragment maturation at regions containing repetitive sequences, such as centromeres and telomeres. The authors used chromatin immunoprecipitation approaches coupled to whole-genome DNA sequencing to show that DNA2 is preferentially localized to centromeric alpha-satellite regions. Next, they combined biochemical and cellular approaches to show that both the nuclease and helicase activities of human DNA2 are required to facilitate Okazaki fragment maturation on centromeric DNA. Loss of DNA2 function leads to ATR/ATM checkpoint activation, late-S cell cycle arrest and chromosome segregation defects. Moreover, inhibition of the ATR checkpoint prevents S-phase arrest and forces DNA2 knockout cells to enter mitosis, thereby leading to mitotic catastrophe. Finally, the authors show that the combined inhibition of DNA2 and ATR activity has a synergistic effect on cancer cell survival pointing to a new strategy for cancer cell killing. Collectively, this is an interesting study that proposes a novel function of the DNA2 nuclease/helicase in replication. There are, however, several major concerns that the authors should properly address to strengthen their results and support their conclusions.

Major concerns:

1. The major criticisms of this reviewer is that while the authors provide compelling evidence for

a role of DNA2 at centromeric regions, their conclusion that the observed defects in cell cycle progression and cell survival (when combining DNA with ATR inhibition) is mainly due to loss of DNA2 function in centromeric DNA replication is overstated and not supported by the data. As the authors state in their manuscript, DNA2 has additional roles in DNA replication and DSB repair. Thereby, DNA2 knockout will suppress all these additional DNA2 functions. Separation of function DNA2 mutants that suppress DNA2 function at centromeric regions, while sparing its previously described function in replication and repair, would be necessary to support the authors' conclusion. While generating separation of function mutants is probably a difficult task, the authors should at least revisit their conclusions to mention how the previously reported functions of DNA2 might contribute to the observed phenotypes.

RE: Employing the SMARD assay, we have direct evidence to suggest that DNA2 functions in centromeric DNA replication. We have also performed ChIP-seq of phosphor-ATR and IF-FISH experiments and showed that phosphor-ATR primarily localizes to centromere in response to DNA2 loss. These two additional experiments have strengthened our conclusion that DNA2's function in centromeric DNA replication is important for cell cycle progression and cell survival. In the Discussion, we have re-visited all of the cellular functions of DNA2 in DNA replication and repair, which are also important for cell cycle progression and cell survival.

2. Figure 3B and 3C. The authors should include immunoblots of total protein levels, in addition to chromatin-bound fractions, to rule out the possibility that DNA2 loss affects CENPA expression.

RE: As suggested, we have checked the level of CENP-A from whole cell lysates (Figure S2B). No significant change of CENP-A was found.

3. The authors should repeat all their cell cycle analyses by co-staining S-phase cells and DNA content because staining DNA content using PI alone does not allow to distinguish between late S and G2/M phases of the cell cycle. Also, the authors should quantify the Sub-G1 fraction in order to test whether these cells are permanently arrested or have undergone cell death (and therefore go to the Sub-G1 fraction).

RE: We claim that the functional deficiency of DNA2 in centromeric DNA replication contributes to cell cycle arrest in late S and G2/M phases. To keep the manuscript focused, we did not pursue any experiments to distinguish the cell cycle arrest between late S and G2/M phases. The quantification of cells in different cell cycle phases is presented in the current Figure S2A, which clearly shows that a marked accumulation of DNA2 null cells in late-S, G2/M, and Sub-G1.

4. Figure S2. As mentioned above, the authors should co-stain S-phase cells and DNA content (BrdU-PI) to better distinguish between late S and G2/M phases of the cell cycle. Moreover,

Figure S2C does not show any significant difference between the cell cycle profiles of DNA2 WT and DNA2 knockout cells. This cell cycle analysis should be repeated at a time point where DNA2 loss leads to a more marked G2/M phase arrest (for example 56 or 72 hours, as shown in Figure S2A).

RE: While we did not pursue the experiment to distinguish late-S from G2/M arrest, we have repeated the experiment with cells that were treated with 4-OHT for 72hr as suggested by the reviewer. Following the treatment, DNA2 null cells accumulate in late-S and G2/M as shown in the current Figure S2A.

5. The authors state that DNA2-null cells, but not WT cells, accumulate in late S/G2 phase. However, Figure S2A shows only the quantification of cells in G2/M without quantifying the percentage of cells in late S/G2.

RE: Since cell cycle profile analysis cannot distinguish if cells are in late S or G2/M, we have revised the previous claim that the DNA2 null cells were arrested in late-S rather than G2/M. Instead, we state in the revised manuscript that the DNA2 null cells are arrested in late S and/or G2/M phase.

6. Figure 4C. Why does treatment with RO-3306 lead to a decrease in DNA2 protein levels? The authors should use nocodazole as an alternative approach to accumulate cells in G2/M and test whether they obtain reproducible results.

RE: We used Nocodazole as an alternative approach to accumulate cells in G2/M phase as shown in Figure S2B. The results are reproducible. There was a decrease of DNA2 when cells were treated with either RO-3306 or Nocodazole, which may be due to cell cycle changes. It is common for the protein levels of enzymes such as FEN1, which is involved in replication, to significantly decrease when cells exit S-phase (Guo et al., 2012, Molecular Cell).

7. To explain the molecular basis for ATR and ATM checkpoint activation observed upon DNA2 loss, the authors state ATR activation may result from the inability of DNA2 to resolve centromeric DNA replication intermediates whereas "ATM is constantly activated by endogenous double strand breaks resulting from defects in DNA2-mediated end resection". What is the actual evidence for this conclusion? How frequent are DSBs in DNA2-depleted cells? The authors should compare the amount of DSBs in DNA2Flox/+/+ versus DNA2Flox/-/- cells to support their conclusion.

RE: We agree with the reviewer that we do not have evidence showing a higher amount of DSBs in the DNA2Flox/-/- cells, and thus the manuscript was revised. The status of ATM checkpoint activation is out of the scope of the revised manuscript. Therefore we did not pursue

measurement of the amount of DSBs in these cells.

8. The authors state on page 13 that "a small population of DNA2-null cells spontaneously entered mitosis". This conclusion is based on the observation that DNA2 loss causes a reduction of phospho-histone 3 positive cells from 1.5% to 0.48%. However, these numbers are very small and the observed differences could be simply associated to small differences in the density of cells in the wells or in the timing of seeding or trypsinization.

RE: The reduction of mitotic DNA2 null cells (phosphor-H3) from 1.5% to 0.48% by flow cytometry is a very consistent observation. We repeated this experiments four times and our p-value is 0.005406 (student's t-test). This result is also supported by our immunofluorescence microscopy data (current Figure 5A and 5B). Nocodazole treatment does not increase the percentage of DNA2 null cells that are in mitotic phase, suggesting that these cells do not enter mitosis.

9. Figure 6A. The authors should provide the same histograms including the Sub-G1 fractions to rule out the possibility that the observed decrease in late S/G2 phase of the cell cycle is due to increased cell death.

RE: The quantification of DNA2-null cells including the sub-G1 cells is now presented in Figure S2A, which excludes a possibility that the observed decrease in late S/G2 phase of the cell cycle is due to increased cell death.

10. Figure 6C. The authors should include better images for the metaphase-spreads. In particular, the metaphase-spreads of the DNA2Flox^{-/-} cells without treatment already show some significant defects compared to control as the chromosomes seem to be more condensed. Is this because of the low picture quality or is a real phenotype? In addition, the authors should use centromere specific probes to investigate whether there are any specific defects at centromeres on the metaphase-spreads.

RE: We have revised this part of the manuscript because this mitotic catastrophe phenotype cannot be used as direct evidence to support our conclusion that centromere is defective in the DNA2-null cells. Mitotic catastrophe may also be due to DNA2's role in RNA primer removal or DSB repair. In this revision, we did high resolution single molecule analysis of replicated DNA, which clearly shows centromeric DNA replication defects in the DNA2-null cells (Figure 3). In addition, centromeric DNA replication defect is also supported by our finding that phosphor-ATR primarily localizes to centromere regions by using ChIP-seq and IF-FISH (Figure 6).

11. Figure 6C. The authors should include in the same figure the data obtained with DNA2Flox^{-/-} cells without inhibitor treatment (4-OHT + ; ATMi/ATRi -).

RE: This part is omitted because this mitotic catastrophe phenotype cannot be used as direct evidence to support our conclusion that centromere is defective in the DNA2-null cells.

Referee #2:

In this manuscript, Li et al. describe the role of DNA2 role on centromeric DNA replication and stability. In general, the manuscript is interesting especially for what concerns the first part in which the authors suggest that DNA2 is binding centromeric DNA, promoting its replication. To support these claims the authors show that that in vitro DNA2 is specifically involved in resolving hairpin structures potentially arising in centromeric DNA. Furthermore, the authors show that DNA2 depletion in cells lead to incomplete centromeric DNA replication, activation of the DNA damage response and mitotic entry failure. Although the first part concerning the involvement of DNA2 in centromere stability is acceptable, the experiments related to consequence of DNA2 deficiency on cell cycle progression are quite disappointing and make little sense considering the role proposed for DNA2 at centromeres. This second part is indeed all based on minimal differences of pH3S10 and blots for phospho Chk1 and Chk2 of poor quality and not well controlled. This unfortunately prevents me to give a positive evaluation of this work.

RE: We have heavily revised the second part of the manuscript. We have done high resolution SMARD assays both with the DNA2 WT and null mutant cells, which clearly shows centromeric DNA replication stalling in the DNA2-null cells (Figure 3). In addition, centromeric DNA replication defect is also supported by our finding that phosphor-ATR primarily localizes to centromere regions by using CHIP-seq and IF-FISH (Figure 6). We hope that the amended experiments addressed the major concerns of reviewer #2.

Critiques

-The experiments presented in the first paragraph only show that DNA2 binds centromeric DNA and do not directly prove that it is required for its replication. Direct evidence for DNA2 involvement in centromeric replication is missing. The Chip-seq experiment should be carried out also on BrdU positive strand to address whether the DNA2 is also present on replicated sequences.

RE: We now have more direct evidence showing DNA2 involvement in centromeric DNA replication. We established a single molecule analysis of replicated DNA to specifically study

the replication status of the centromeric DNA in the DNA2 null cells. In Figure 3, our data clearly shows a defect of fork progression at the centromere but not at the non-centromeric DNA. In addition, centromeric DNA replication defect is also supported by our finding that phosphor-ATR primarily localizes to centromere regions by using CHIP-seq and IF-FISH (Figure 6).

-It is not clear whether the observations related to under-replication of centromeric DNA are due to the absence of DNA2 action on hairpins or to checkpoint activation. ATRi should be used to clarify this issue.

RE: To clarify the relationship among DNA2 deficiency, defects in centromere DNA replication, and ATR activation, we conducted SMARD, IF-FISH and ChIP-seq on phosphor-ATR in WT and DNA2 knockout cells as well as ATRi as suggested by the reviewer. The new data clearly showed that absence of DNA2 at centromere resulted in stalled replication forks at centromeres and ATR activation. Because ATR under normal condition is not activated at centromeres (Aze et al. *Nature Cell Biology* 18, 684–691, 2016), we suggested stalled replication forks or defects in processing of secondary structures at centromeres is the cause of ATR activation. Consistently, ATRi on its own did not rescue arrest of DNA2 knockout cells at late-S and G2/M phase (Figure S7).

- Binding of DNA2 to centromeric regions is not supported by alternative approaches such as colocalization with centromere markers by immunofluorescence (using for example anti CENP-A antibodies or fish probes for satellite DNA).

RE: DNA2 expression is low in the HCT-116 cells, therefore, we were not able to demonstrate the co-localization between DNA2 and CENP-A. We were able to use CHIP-seq, which has high sensitivity and resolution to show that DNA2 binds to centromere. In addition, we were able to do similar experiments for co-localization between phosphor-ATR and CENP-B box to support the overall conclusion.

-The consensus sequence for the major repeats found at centromere is not novel. The authors did not cite important works, which clearly identified a similar consensus sequence underlying the formation of stem-loop structures (Grady D. et al, *PNAS* 1992, 89, 1695-1699; Catasti P. et al *Biochemistry*, 1994, 33, 3819-3830). This somehow diminishes the novelty of these findings.

RE: Our data shows that DNA2 specifically binds to typical centromere motifs that were identified previously. We do not suggest that we have identified any new motifs of centromeric DNA. We have now cited these two important publications.

-Some of the repeats resemble telomeric sequences. Since DNA2 binds also TRF1 (Lin *et al.*, *Embo J.* 2013), can the authors exclude that the sequence belongs to telomeric regions instead than to centromeres?

RE: We agree that telomeric DNA is expected to be in the DNA pool (Lin *et al.* *EMBO J.* 32(10):1425-39, 2013). Technically, telomeric DNA was excluded in any reference genome so that we would be able to do better alignment analyses with centromeric DNA. This is because different cell lines have different telomere length and it changes when cell status is different.

-It is not clear how the fold enrichment is calculated in Fig 1c.

RE: The relative enrichment of centromere peaks is evaluated by comparing two fractions, one is dividing the number of bases in the peaks within the centromere region by the total bases in the centromere regions, and the other is dividing the peaks not in the centromere region by the total bases in the genome that are not in the centromere region. This ratio represents whether peaks are enriched in the centromere region. P value is calculated by Fisher's exact test. The information has been included in the Methods.

-There are a number of inconsistencies in figure 3-4 and related panels.

First, in S2 figure (panel A and B), the authors show that after depletion of DNA2 the intra-S-phase checkpoint is activated and as evidence they show Chk1 phosphorylation levels. However, there is no accumulation in S-phase in the FACS profiles shown.

RE: The cell cycle arrest is far slower than checkpoint signaling. The activation of Chk1 would need to reach a threshold to be able to arrest the cell cycle. Therefore it makes sense that cell cycle arrest may happen later than the Chk1 phosphorylation.

Second, the authors suggest that Chk1 activation is due to centromeric defects arising in the absence of DNA2. However, this claim makes no sense as DNA2 is known to act in many other genomic regions, being required for the processing of double strand breaks and long flaps of Okazaki fragments during DNA replication. Therefore, activation of Chk1 could well be due to lack of functional DNA2 in different parts of the genome.

RE: We have now developed single molecule analysis of replicated DNA and show clearly that centromeric DNA replication is stalled (Figure 3). We also collected direct evidence showing that without DNA2 to process centromeric DNA replication, phosphor-ATR clearly localizes to centromeric DNA by the CHIP-seq (Figure 6A), and forms clear foci at centromere by IF-FISH experiments (Figure 6C). We do not exclude the possibility that activation of Chk1 is due to lack

of DNA2's role in a different mechanism of genome maintenance and have discussed these in the revised manuscript.

Third, the effects on CENP-A loading (Fig S3 panel B) could be due to the accumulation of cells in G2/M, rather than under-replication of centromeric regions.

RE: To rule out this possibility, we synchronized cells in G2/M phase by two different methods and show that G2/M cells have similar levels of CENP-A on the chromatin, which also is in agreement with the literature (Foltz, D.R. et al. *Cell* 137, 472-84, 2009).

Fourth, the authors should improve the quality of the blots, which are rather poor.

RE: All the western blotting was repeated. High quality blots from a single gel were presented in Figure 6A, Figure S2B and S2C, and Figure S6B.

-I disagree with the interpretation of Figure 4B and 4C as they do not provide enough evidence to claim that there is a block in S-phase. Cells seem to proceed slowly in S-phase and then progress into G2. Increasing levels of CyclinB and decreasing levels of CyclinE are consistent with this different interpretation.

RE: We did not attempt to distinguish late-S from G2/M arrest since it is not a focus of this paper. This part has been revised.

-Blots are in Fig4C and S2B are only done with DNA2^{-/-} cells. They should be performed with DNA WT cells, which also show high signal for phosphoChk1 (Fig 6B)

RE: A new experiment was conducted. This revised data is now in Figure S6 and S7.

-In the figure 4A, the authors should quantify the mitotic entry capacity of DNA2 null cells by FACS analysis instead of Immunofluorescence

RE: We did FACS analysis as suggested. Its quantification is presented in Figure 5D.

-Immunofluorescence images are of extremely poor quality, which does not seem to be due to file compression. Better quality images should be provided.

RE: High quality images are now provided in Figure 5A, Figure 6C, and Figures S3-S6.

-In Fig 5C the authors calculate a congressional index. This parameter is not informative. They should instead calculate the percentage of cells that have all the chromosomes aligned on the metaphase plate as it is usually done. They should also specify that they are monitoring cells

after nocodazole release (this is only mentioned in materials and methods). It would be worth adding MG132 to block anaphase before calculating the percentage of cells with aligned chromosomes as congression delay could be simply due to poor recovery from nocodazole mediated arrest.

RE: This experiment was done by blocking the cells at G2/M border with RO-3306 and then released in warm medium for 45-60min. No nocodazole was used in this experiment as it will affect the formation of the metaphase plate by itself. In Figure 5F, we have calculated both percentages of cells that have all the chromosomes aligned on the metaphase plate (normal) and of cells with misaligned chromosomes (misaligned). The anaphase cells with chromosome mis-segregation or lagging chromosomes were also counted in Figure S5. In addition, we have calculated the congression index (Figure 5G) as defined by Green RA, Kaplan KB (J Cell Biol 2003; 163: 949–961). The ratio between the width (parallel to the spindle poles) and length (perpendicular to the spindle poles) of the metaphase chromosome mass was utilized to calculate the chromosome congression index. Due to centromere under-replication in DNA2 null cells, an impaired chromosome alignment widens the metaphase plate (congression index).

Figure 6:

Panel A is confusing. The blots in panel B are clearly run on different gels. This should be specified. Are all the samples from DNA2 null cells? Looking at the WB of DNA2 it is not possible to assess the presence-absence of DNA2. In panel C a graph should be added. Also the authors should clarify how they measured mitotic catastrophe.

RE: New western blotting experiments were performed. High quality data are presented in the current Figures S2 and S6. As mentioned above, the mitotic catastrophe part is omitted in the revised manuscript.

Figure 7:

Panel A shows the inhibition of DNA2 in Breast cancer cells. Could the authors justify the lack of any cell cycle block in the presence of active Chk1 and Chk2 proteins after the addition of DNA2 inhibitor (Panel B)? Also, these experiments should be repeated using a panel of cancer cell lines instead of one cell line. DMSO treated cells should be included as control. Again, the blots shown in panel B seem to be run on different gels as separate experiments.

RE: Our understanding is that cell cycle block is a consequence of checkpoint activation and happens later than signaling. Checkpoint activation must pass a threshold to arrest cell cycle. Therefore, in some cases we observed checkpoint activation but not cell cycle arrest.

Three different cell lines (Figure 7) including MCF7 (breast cancer), and HCT-116 (colorectal cancer) and H460 (non-small cell lung cancer) have been tested in this revision. Similar results were observed in these three cell lines. Vehicle control was used in all of our experimnts. For the last question, the blots were from a single gel unless specified.

Referee #3:

Li and colleagues outline work in this manuscript that the authors conclude shows that hDNA2 is responsible for centromeric DNA replication. They base their conclusion on cellular sequencing results that show high occupancy of DNA2 at centromeres. This observation is followed up by mechanistic studies wherein the authors used random and centromeric DNA substrate mimics to assess the activity of Dna2 on these substrates. Additional cellular results show that the ATM/ATR pathway is activated due to incomplete centromeric replication. The fundamental conclusion of the study is that since DNA2 is important for centromeric replication, it can be used as a chemotherapeutic target along with ATR inhibitors in the treatment of cancer. There are considerable issues with the interpretation of results in this study as presented. Primarily, centromeric DNA requires the help of MMR proteins to help resolve the looped structures. Hence choice of the HCT116 cell line (which are deficient in MLH) is an odd choice for these studies.

RE: We have compared the involvement of DNA2 in centromere function in MMR-proficient and MMR-deficient cells as shown in the current Figure S2C. Our results indicated that the role of DNA2 in maintaining centromere integrity is not affected by MMR status.

Secondly, there is no information about how the sequencing libraries were made in the materials and methods or how many times replicates were done in the sequencing analysis? What was the read size, and was this paired-end sequencing? Was the chromatin fragmented by sonication or micrococcal nuclease digestion? Also for the sequencing data that is presented in the study, was it merged data from multiple samples or from a single experiment? How reproducible were the results? Data from individual sequencing needs to be added in the supplementary results to show reproducibility. The data with DNA2 flox^{-/-} cells and response of the repair pathways is not completely novel, with even the authors acknowledging their data is consistent with previous findings. The results presented in this manuscript remain purely observational without taking into consideration the confounding effects of the role of DNA2 replication of the whole genome.

RE: The methods for preparation of the sequencing libraries was described to our best abilities in the “High-throughput sequencing data analysis”, “Immunocapture and isolation of BrdU-labeled DNA”, and “Chromatin immunoprecipitation (ChIP)” of the Methods sections. These sections covered the questions asked here. All of our experiments were confirmed and representative results were shown as mentioned in our description in the Method. Our new experiments using single molecule analysis of replicated DNA showed replication fork stalling in the centromeric DNA but not the non-centromeric regions (current Figure 3). In addition, our phosphor-ATR CHIP-seq and IF-FISH (current Figure 6) clearly show that the centromere is the major region that has replication-stalling signaling when DNA2 is knocked out.

Major Concerns

Figure 1: While it is hard to judge the sequencing data without having any information on the

exact methodology used to generate the data or its reproducibility, there are other considerations for the observation that need to be either analyzed or discussed. For example, the conclusion from the sequencing data that DNA2 was involved in centromeric replication is solely based on higher occupancy of DNA2 compared to FEN1. While FEN1 cannot resolve fold-back intermediates it can do so in the presence of other helicases, especially the 5'-3' RNA/DNA helicase, Pif1. Were other proteins involved in the Okazaki fragment maturation considered in the study, for example, Pif1, WRN and RPA?

RE: We have developed single molecule analysis of replicated DNA to assess the replication status of centromeric DNA (current Figure 3). Our data demonstrated stalled replication forks at the centromeric DNA but not in the bulk of the genome in absence of DNA2. Examination of the role of FEN1 and its associated protein will be our future work.

Fig S1: How efficient was the depletion experiment? Are the authors confident about not having any contaminations in the depleted pool of cells?

RE: The WB blots show a clear depletion of DNA2 (Figures S2B, S6B, and 4B). In addition, 79% of the DNA2 null cells can be arrested at G2/M (Figure S2A). Both pieces of the data suggest that there is little or no contamination in the depleted pool of cells.

Figure 2: Templates having the 5' fold required ~15-20 fold excess of Dna2 to resolve the structure. While this was mentioned in the materials and methods section, it should be included in the actual results, so that the reader can immediately grasp the requirement for higher concentration of protein for fold back flap resolution. Interestingly, the wild type protein did not show any cleavage below the 5' terminal product, suggesting the helicase activity of Dna2 was not sufficiently strong enough to create a ligatable nick. Does RPA aid in this function of Dna2 by melting out some of the structure. Based on data from Aze et al, 2016, RPA abundance was lower at centromeric DNA hence assessment of abundance of RPA at the centromere in this system would have been helpful for the in vitro study.

RE: The concentration of purified DNA2 is specified in the Figure legend for reader to grasp the requirement for higher concentration to resolve fold back structures. It has been documented that DNA2 does not cleave the 5' terminal product but works on longer flaps (Bae *et al.* Nature. 412(6845):456-61, 2001). Without RPA, DNA2 alone can process replication intermediates to facilitate DNA replication (Lin *et al.* EMBO J. 32(10):1425-39, 2013). Our future work will be to identify additional components that make this reaction highly efficient.

The cleavage products should contain quantitation. It is intriguing that the 5' and 3' labeled bubbled centromeric substrates were resolved with different efficiencies by the wild-type enzyme, though it should have been the same.

RE: The current experiment was designed to detect the nature of the reaction, instead of measuring enzyme kinetics and to compare the activities between WT and mutant enzymes. Our results give a clear qualitative conclusion that DNA2 mutants do not resolve fold back DNA secondary structures due to lack of either nuclease or helicase activity (Figure 2D and 2E). The seemed difference in enzyme activity efficiency with differently labeled substrates is probably caused by the reaction conditions.

Did the authors try test ligation of the resolved fold back flap substrate to ensure DNA2 could process these structures in the absence of FEN1?

RE: It has been demonstrated that RPA suppressed the DNA2 activity against a short flap, a product from the DNA2 action both on simple long flap and stem-loop structured substrate (Bae et al., Nature, 2001). Therefore, we don't expect that DNA2 will be able to generate a ligatable product in the absence of FEN1.

Figure 3: What is the consequence of overexpression of FEN1 in the DNA2-null cells?

RE: In yeast, it has been shown that overproduction of yFEN1 suppresses the temperature-sensitive growth of *dna2* mutants (BUDD *et al.* JBC, 2136–2142, 1997).

Figure 4-5: Arrest of DNA2 cells in the S or G2 phase and activation of the ATR/ATM pathways could be the consequence of its role in nuclear Okazaki fragment maturation and may not necessarily only be caused due to delays in centromeric DNA replication

RE: Our new experiments using phosphor-ATR ChIP-seq and IF-FISH (Figure 6) clearly show that the centromere is a region where replication-stalling signaling is activated when DNA2 is knocked out. We also have new direct evidence showing that centromeric DNA replication is indeed stalled in DNA2 null cells (Figure 3). We do not exclude Okazaki fragment maturation as a resource for ATR/ATM activation. Other possible causes of ATR pathway activation has been discussed in the Discussion of the revised manuscript.

Thank you for submitting a new version of your manuscript on DNA2 in centromeric replication. It has now been reviewed once more by all three original referees, whose comments are copied below. I am pleased to inform you that all three referees consider the study substantially improved and now in principle suitable for publication in The EMBO Journal. Nevertheless, referees 1 and 2 still have several concerns regarding the experimental data and their interpretation. Should you be able to satisfactorily address/clarify these remaining issues through a final round of revision, we should be happy to ultimately offer publication of a re-revised manuscript in The EMBO Journal.

 REFEREE REPORTS

Referee #1:

The authors included new data to support their finding that the human DNA2 helicase/nuclease plays an important role in centromeric DNA replication. In particular, they established an elegant single-molecule DNA fiber assay to directly monitor replication perturbations at centromeric regions and investigate the role of DNA2 in this process. The revised version of the manuscript is significantly improved and the authors properly addressed most of my previous comments. There are, however, few remaining concerns that the authors should address to support their conclusions and warrant publication in the EMBO journal.

Major points:

1. Page 8. The authors' conclusion that "the DNA2 helicase activity first separates the stem to make a single ssDNA that is long enough (~10 nts) for nuclease cleavage; then, when the ssDNA at the junction is exposed, DNA2 cleaves the whole hairpin structure and removes the structured DNA at the replication fork" is overstated and not fully supported by the data. The authors show that the combined helicase and nuclease activities of DNA2 are indeed required to cleave these hairpin structures. However, they do not provide any direct evidence that the helicase activity of DNA2 is required to resolve the hairpin structure before nuclease cleavage.
2. The authors should repeat the SMARD assay using DNA2-null cells complemented with nuclease- or helicase-dead DNA2 to further support their conclusion that both activities are critical for centromere maintenance.
3. Page 12. The authors state that "The loss of mitotic entry and subsequent cell death occurred because of cell cycle arrest in late S/G2 phase, which are consistent with previous observations". Where are the data on cell death?
4. Page 13: The authors state that "ATR inhibition on its own did not rescue arrest of DNA2 knockout cells at late-S and G2/M phase (Figure S7). Thus, these results support a model wherein the DNA2-null cells are arrested in late S/G2 phase at least in part by DNA damage checkpoint activation." If ATR inhibition does not alter cell cycle distribution/entry in mitosis, how can authors conclude that the DNA2-null cells are arrested in late S/G2 phase by DNA damage checkpoint activation? Along the same line, if ATR inhibition is not alleviating checkpoint arrest and entry in mitosis, what is the mechanism of ATRi-induced cell death in DNA2 inhibited cancer cells?

Minor points:

1. The authors should use "CENPA" or "CENP-A" throughout the manuscript for consistency.
2. Figure 3B and 3C. The mean values should be replaced with the median values for accuracy.
3. Figure 4C is not cited anywhere in the text.
4. Figure 7A. The authors should include the actual concentrations of the ATR and DNA2

inhibitors.

5. Page 8. "Figure 1B" should be corrected with "Figure 2B" at the end of the sentence that starts with "When incubated with a model DNA substrate mimicking the hairpin DNA structure...")

6. The authors should clearly explain which is the difference between Figure 2D and Figure 2E in the main text and in the Figure legend.

7. Page 14 (first paragraph of the discussion). The author should remove the reference of Thangavel et al., at the end of the sentence "impact of secondary structures on these regions can be easily overlooked..." because the paper of Thangavel et al does not focus on the analysis of secondary structures.

8. Page 16. "A recent studies" should be corrected with "A recent study" and a period should be included before the sentence.

Referee #2:

The authors in this manuscript highlight an important role for DNA2 in centromere replication. The paper describes exciting findings consolidating the emerging concept that centromere replication poses a major hurdle to the replication machinery. In the revised version, the authors have made a significant effort to address my previous critiques. They have tried to pin the function of hDNA2 at the centromere by setting up assays to study the function of this protein in centromeric DNA replication. Thanks to these new results the story now stands on more solid ground. However, some issues still remain and should be addressed. Here my comments:

Major Critiques:

-The SMARD assay applied to centromere is a useful tool to specifically assess centromere replication. However, it is not clear what is the point the authors want to make. It is indeed unclear whether the assay shows an overall problem in DNA replication initiation or fork progression (or both). This should be clearly indicated.

-Double labelling (IdU and CldU) in fiber assays is used to reveal fork progression issues such fork asymmetry. From the examples shown it is very hard to determine whether the shorter green tracts correspond to impaired fork progression as there is no evidence of replication initiation (no red signal on the DNA2 depleted tracts). From what it is shown in fig 2B in DNA2 null samples it looks that there is no red and green labelling and forks never initiated. Is this the case? If so, the authors should consider the possibility that DNA2 defects lead to ATR activation that inhibits replication origin firing, which has been shown to prevent replication onset and not just fork progression (Aze et al 2016 NCB). If replication impairment was due only to fork progression impairment there should have been evidence of impaired fork progression such as asymmetric green tract labelling, but this is not shown.

-The timing of the labelling is also quite unusual and not very informative. Authors should specify why they use a 4+4 hours IdU-CldU labeling protocol in the SMARD assays as most of other fiber analyses are done with much shorter pulses, which allow to identify origin firing and fork progression issues. In 4 hours most of the fibers should be all red with limited green tracts. Is this long labelling protocol needed to catch the delayed kinetic of centromeric DNA replication with respect of S-Phase onset? If so the authors could try to synchronize their cells in S-phase or at least explain the rationale of their strategy.

-The authors should quantify the percentage of centromeric fibers over the non centromeric ones in the centromeric SMARD assay to have a more precise quantification of the impact of DNA2 deletion on overall DNA replication. Without these data, it is difficult to determine the fraction of unreplicated/partially replicated DNA caused by the absence of DNA2.

-The pATR results shown in Fig 6 are impressive and novel. However, there is no mechanism to support these findings. The authors should at least show that RPA is accumulated on the centromeric foci with increased pATR signal.

As negative control the authors could show that other agents that cannot activate ATR at centromeres such as Aphidicolin as shown in Aze et al 2016 do not induce ATR activation and RPA accumulation, in contrast to DNA2 deletion, which by disrupting normal fork structures, would allow RPA loading and ATR activation.

This experiment would consolidate the parallel with published evidence, which showed the ATR at centromere is only activated when there is a loss of replication fork integrity (induced by Topoisomerase inhibitor CPT as in Aze et al 2016).

-The CPT experiment presented in Figure S6A is not described in the text.

-If the authors want to compare the effects of CTP and DNA2 on ATR activation, they need to show that DNA2 loss and CPT have a similar mechanism of action (e.g. Is pATR activation by CPT peaking at centromeres? Is CPT inducing RPA accumulation? Do CPT and Aphidicolin have different effects on centromere pATR activation?).

Minor corrections

-Aze et al showed that ATR Inhibition and not activation is important for centromere replication. Therefore, it is better to substitute the sentence on Page 13 "ATR activation is important for ensuring a sufficient time for DNA replication machinery to deal with centromeric replication problem (Aze et al, 2016)" with "Modulation of ATR activity is important for ensuring a sufficient time for DNA replication machinery to deal with centromeric replication problem (Aze et al, 2016).

-On page 14 it should be "we modified the SMARD technology to specifically" as SMARD has been already developed for telomere replication detection.

-On page 13 "Costanzo et al 2016" should be substituted with Aze et al 2016

-On page 16 the authors should properly cite the existing literature. They write: "A recent studies observed extended ssDNA regions at centromeric DNA regions, especially under external stresses (Aze et al, 2016). The origin and function of these extended ssDNA regions are not clear" The sentence should be rewritten and should incorporate the concepts described below. Centromere intrinsic topological structure kept together by CENPA and SMC2-4 prevents extensive ssDNA accumulation in unchallenged conditions. Centromere were shown to accumulate ssDNA in response to fork stalling only when the centromere structure was disrupted.

Title: I think it should be revised. I suggest some alternative titles: "hDNA2 promotes centromeric DNA replication and genome stability"; "hDNA2 ensures genome stability by promoting centromeric DNA replication"

For the discussion the authors could mention that several repair proteins were shown to accumulate on centromeric DNA, which might also be important for centromeric DNA stability and replication.

Referee #3:

In the extensively revised manuscript Li and colleagues provide additional evidence that suggests that DNA2 plays a role in centromeric replication. Experiments performed as suggested by the reviewers and the development of the SMARD technique has significantly enhanced support of the

conclusions of the manuscript. All previous concerns of this reviewer have been satisfactorily addressed in the revised manuscript.

2nd Revision - authors' response

7 March 2018

Point-by-point response to the reviewers' comments

[Introduction] We would like to thank the reviewers for their support to publish the work in EMBO Journal and constructive suggestions and advice to improve the quality of the current manuscript. We have addressed the following major points of the reviewers with additional experiments:

Reviewer #1:

1. We have provided additional evidence to support that the helicase activity of DNA2 is required to resolve the hairpin structure before nuclease cleavage (new Figure EV3).
2. We have repeated the SMARD assay using DNA2-null cells complemented with nuclease- or helicase-dead DNA2 to further support the conclusion that both activities are critical for centromere maintenance (new Figure 3B and 3C).
3. We have generated Figure EV4E showing quantification of cell death in DNA2 null mutant cells (new Figure EV4E).
4. We have performed additional experiments (new Figure 7F-7I) to strengthen the point that the DNA2-null cells are arrested in late S/G2 phase by DNA damage checkpoint activation and explained the mechanism of ATRi-induced cell death in DNA2 inhibited cancer cells.

Reviewer #2:

5. We have explained the strategy for the DNA replication track labeling in the SMARD assay (new Figure 3B and 3C).
6. We re-analyzed the SMARD data (new Figure 3B and 3C).
7. We have shown that RPA accumulates on the centromeric foci with increased pATR signal and included aphidicolin treatment as a negative control (new Figure 6C and 6D).

We have also clarified all other issues that the reviewers have raised and made corrections based on their minor points, including changing the title of the manuscript. All the changes in the text and figures are marked with a vertical red line corresponding to the page numbers included in this document. The following are the point-by-point response to the reviewers' comments.

Referee #1

The authors included new data to support their finding that the human DNA2 helicase/nuclease plays an important role in centromeric DNA replication. In particular, they established an elegant single-molecule DNA fiber assay to directly monitor replication perturbations at centromeric regions and investigate the role of DNA2 in this process. The revised version of the manuscript is significantly improved and the authors properly addressed most of my previous comments. There are, however, few remaining concerns that the authors should address to support their conclusions and warrant publication in the EMBO journal.

Major points

1. Page 8. The authors' conclusion that "the DNA2 helicase activity first separates the stem to make a single ssDNA that is long enough (~10 nts) for nuclease cleavage; then, when the ssDNA at the junction is exposed, DNA2 cleaves the whole hairpin structure and removes the

structured DNA at the replication fork" is overstated and not fully supported by the data. The authors show that the combined helicase and nuclease activities of DNA2 are indeed required to cleave these hairpin structures. However, they do not provide any direct evidence that the helicase activity of DNA2 is required to resolve the hairpin structure before nuclease cleavage.

[RESPONSE] We have included additional evidence in Figure EV3 to support our statement on the DNA2 helicase/nuclease-mediated stem/loop processing. We designed DNA stem/loop structures that contained one, two, or three stem/loop structures and comprehensively mapped the cleavage sites based on the product sizes shown on the gel images. Our cleavage site mapping indicated that DNA2 nuclease activity only works on the ssDNA regions. The stems are separated by the helicase activity of DNA2, which has been demonstrated in multiple previous publications (Lin et al., 2013, EMBO J., Ronchi et al., 2013, AJHG; Liu et al., 2016, eBioMedicine). Additional information is included in the figure legends for Figure EV3 and in the manuscript text (Pages 8-9).

2. The authors should repeat the SMARD assay using DNA2-null cells complemented with nuclease- or helicase-dead DNA2 to further support their conclusion that both activities are critical for centromere maintenance.

[RESPONSE] We have followed the suggestion by the reviewer and complemented the DNA2-null cells with nuclease- or helicase-dead DNA2. Our results showed that elimination of either nuclease or helicase activity leads to stalling of the DNA replication fork specifically in centromere regions, similar to the observation made with the DNA2 null mutant (see revised Figure 3). The new data further support our statement that both the nuclease and helicase activities are required for resolution of the stem-loop structures and for facilitating DNA replication progression in the centromere regions (Pages 9-11).

3. Page 12. The authors state that "The loss of mitotic entry and subsequent cell death occurred because of cell cycle arrest in late S/G2 phase, which are consistent with previous observations". Where are the data on cell death?

[RESPONSE] We have now presented the cell death data in Figure EV4E, which shows a larger sub-G1 apoptotic cell population for the DNA2 null mutant as compared to the WT (Page 13).

4. Page 13: The authors state that "ATR inhibition on its own did not rescue arrest of DNA2 knockout cells at late-S and G2/M phase (Figure EV7). Thus, these results support a model wherein the DNA2-null cells are arrested in late S/G2 phase at least in part by DNA damage checkpoint activation." If ATR inhibition does not alter cell cycle distribution/entry in mitosis, how can authors conclude that the DNA2-null cells are arrested in late S/G2 phase by DNA damage checkpoint activation? Along the same line, if ATR inhibition is not alleviating checkpoint arrest and entry in mitosis, what is the mechanism of ATRi-induced cell death in DNA2 inhibited cancer cells?

[RESPONSE] ATR inhibition did not erase the checkpoint or rescue late S/G2 arrest. Instead, it further enhanced S/G2 arrest and apoptosis. This is the foundation for testing if ATR and DNA2 inhibitors had synergistic effects in killing cancer cells. To strengthen the point, we have added four panels in Figure 7 (7F-7I), where we show the cell cycle profiles after treatment with the DNA2 inhibitor and ATR inhibitor. Combined inhibition of DNA2 and ATR further arrests cells at late-S/G2 phase and decreases the number of cells that enter the mitotic phase (Figure 7F -7I and Figure EV6). One of the possible mechanisms is that ATR inhibition destabilizes stalled replication forks and causes replication catastrophe (Buisson et al. Mol Cell. 2015). Additional

data/interpretations (Figure 7F-7I) and references are included in the text, figure legends and references (Page 15).

Minor points

1. The authors should use "CENPA" or "CENP-A" throughout the manuscript for consistency.

[RESPONSE] Corrections are made for consistency.

2. Figure 3B and 3C. The mean values should be replaced with the median values for accuracy.

[RESPONSE] Mean values have been replaced by the median values.

3. Figure 4C is not cited anywhere in the text

[RESPONSE] Figure 4C now is cited in the text (Page 12)

4. Figure 7A. The authors should include the actual concentrations of the ATR and DNA2 inhibitors.

[RESPONSE] The actual concentrations of the ATR and DNA2 inhibitors are now included in the figure legends (Figure 7).

5. Page 8. "Figure 1B" should be corrected with "Figure 2B" at the end of the sentence that starts with "When incubated with a model DNA substrate mimicking the hairpin DNA structure..."

[RESPONSE] "Figure 1B" has been corrected to "Figure 2B" (Page 8).

6. The authors should clearly explain the difference between Figure 2D and Figure 2E in the main text and in the Figure legend.

[RESPONSE] The substrates were radio-labeled at 5' end of the flap DNA strand in Figure 2D, while they are radio-labeled at 3' end of the flap DNA strand in Figure 2E. These now have been labeled in the figure and explained in the text (Page 7) and figure legends.

7. Page 14 (first paragraph of the discussion). The author should remove the reference of Thangavel et al., at the end of the sentence "impact of secondary structures on these regions can be easily overlooked..." because the paper of Thangavel et al does not focus on the analysis of secondary structures.

[RESPONSE] The reference of Thangavel et al. in Discussion section has been removed (Page 14).

8. Page 16. "A recent studies" should be corrected with "A recent study" and a period should be included before the sentence.

[RESPONSE] The sentence has been re-written based on the suggestion from Reviewer #2 (Page 18). The grammatical error has been corrected.

Referee #2

The authors in this manuscript highlight an important role for DNA2 in centromere replication. The paper describes exciting findings consolidating the emerging concept that centromere replication poses a major hurdle to the replication machinery. In the revised version, the authors have made a significant effort to address my previous critiques. They have tried to pin the function of hDNA2 at the centromere by setting up assays to study the function of this protein in centromeric DNA replication. Thanks to these new results the story now stands on more solid ground. However, some issues still remain and should be addressed. Here my comments:

Major Critiques

-The SMARD assay applied to centromere is a useful tool to specifically assess centromere replication. However, it is not clear what it is the point the authors want to make. It is indeed unclear whether the assay shows an overall problem in DNA replication initiation or fork progression (or both). This should be clearly indicated.

[RESPONSE] The major point of the current manuscript is to illustrate the role of DNA2 in processing of DNA secondary structures in the centromere regions and in facilitating DNA replication fork progression. However, this did not rule out that DNA2 deficiency might affect DNA replication initiation, possibly due to checkpoint activation. Therefore, following the reviewer's suggestion, we re-analyzed the DNA fibers at centromeres and non-centromeres.

To exclude DNA fibers from non-replicating cells, we counted fibers with IdU/CldU labeling at centromeres only if they showed labeling in the non-centromeric and/or centromeric regions. We interpreted the presence of the red- or green-only tracks as initiation events, and the length and distribution of the green segment of red-green tracks as indicating the rate of replication progression. With that change, the fiber length distribution in the both centromeric and non-centromeric regions in WT cells is bell-shaped, which is similar to previous reports (e.g., Thangavel et al. 2015, JCB). When DNA2 was knocked out, with or without being complemented by the helicase or nuclease dead (ND or HD) mutants, the fiber length distribution in non-centromeric regions did not change significantly from the WT distribution. In contrast, the number of red- or green-only tracks and the lengths and distributions of green segments of red-green tracks at the centromeric region in DNA2 mutant cells were remarkably reduced and altered from those in the WT (Figure 3A-3C). Twenty to thirty percent of DNA fibers even displayed no detectable green or red tracks at the centromeric regions in the DNA2 null, ND, or HD mutant cells, compared to 1% for the WT (Figure 3A and 3C). With respect to red-green tracks, the median length (~40 μm) of green segments at the centromeric region in DNA2 null, ND, or HD cells was remarkably shorter than that (224 μm) in the WT cells (Figure 3C). These findings indicated defects in DNA replication initiation and replication fork progression at the centromeric regions in DNA2 mutant cells as compared to the WT cells (Page 9-11).

-Double labelling (IdU and CldU) in fiber assays is used to reveal fork progression issues such fork asymmetry. From the examples shown it is very hard to determine whether the shorter green tracks correspond to impaired fork progression as there is no evidence of replication initiation (no red signal on the DNA2 depleted tracks). From what it is shown in fig 2B in DNA2 null samples it looks that there is no red and green labelling and forks never initiated. Is this the case? If so, the authors should consider the possibility that DNA2 defects lead to ATR activation that inhibits replication origin firing, which has been shown to prevent replication onset and not

just fork progression (Aze et al 2016 NCB). If replication impairment was due only to fork progression impairment there should have been evidence of impaired fork progression such as asymmetric green tract labelling, but this is not shown.

[RESPONSE] We have carefully considered the reviewer's comments and re-analyzed the DNA fibers in the SMARD assay. In DNA2 null samples, indeed, there are tracks with no red or green labeling but with labels beyond the centromeric regions, indicating forks never initiated in the centromeric regions. Because there was labeling in the non-centromeric regions in the same chromosome, we think that the secondary structures blocked fork initiation and progression in the centromere in the absence of DNA2. It is also likely that ATR activation due to DNA2 defects may inhibit subsequent replication origin firing. We have revised the manuscript to clarify this and rewritten the discussion (Pages 9-11).

-The timing of the labelling is also quite unusual and not very informative. Authors should specify why they use a 4+4 hours IdU-CldU labeling protocol in the SMARD assays as most of other fiber analyses are done with much shorter pulses, which allow to identify origin firing and fork progression issues. In 4 hours most of the fibers should be all red with limited green tracts. Is this long labelling protocol needed to catch the delayed kinetic of centromeric DNA replication with respect of S-Phase onset? If so the authors could try to synchronize their cells in S-phase or at least explain the rationale of their strategy.

[RESPONSE] In the modified SMARD assay, the cells were labeled with iodo-deoxyuridine (IdU) for 4 hours during the first labeling period and with chloro-deoxyuridine (CldU) for another 4 hours during the second labeling period (Norio and Schildkraut, Science, 2001, Demczuk et al. 2012, and Drosopoulos et al. JCB. 2015). These 4 hour labeling periods provide the time required to replicate the DNA regions analyzed, so that we could use the red-green DNA molecules, which are representative of different DNA replication stages, to analyze the initiation and progression of DNA replication forks across the genome (Norio and Schildkraut, Science, 2001). Such a relatively long time period for labeling is particularly important for analyzing the replication of difficult-to-replicate DNA regions using the SMARD assay (Drosopoulos et al. JCB. 2015). (Pages 9-11)

-The authors should quantify the percentage of centromeric fibers over the non centromeric ones in the centromeric SMARD assay to have a more precise quantification of the impact of DNA2 deletion on overall DNA replication. Without these data, it is difficult to determine the fraction of unreplicated/partially replicated DNA caused by the absence of DNA2.

[RESPONSE] The quantification on the percentage of centromeric fibers over the non-centromeric ones in the presence or absence of DNA2 measured by SMARD assay is now presented in Figure 3B (non-centromere) and 3C (centromere). In the current revision, we re-analyzed the SMARD data. We counted fibers with IdU/CldU labeling at centromeres only if there was labeling in the non-centromeric regions in the same chromosomal DNA molecule so that we would exclude DNA fibers from non-replicating cells. With that change, we found that the number of red- or green-only tracks and the lengths and distributions of various green tracks at the non-centromere regions in DNA2 mutant cells (null, ND, and HD) were similar to those in the WT (Figure 3A and 3B), which is in agreement with the previous report (Thangavel et al, 2015). However, the number of red- or green-only tracks and the lengths and distributions of various green tracks at the centromeric region in DNA2 mutant cells were remarkably reduced and altered from those in WT cells (Figure 3A and 3C). The detailed analyses and results are included in the Results section (Pages 9-11).

-The pATR results shown in Fig 6 are impressive and novel. However, there is no mechanism to support these findings. The authors should at least show that RPA is accumulated on the centromeric foci with increased pATR signal.

As negative control the authors could show that other agents that cannot activate ATR at centromeres such as Aphidicolin shown by Aze et al 2016 do not induce ATR activation and RPA accumulation, in contrast to DNA2 deletion, which by disrupting normal fork structures, would allow RPA loading and ATR activation.

This experiment would consolidate the parallel with published evidence, which showed the ATR at centromere is only activated when there is a loss of replication fork integrity (induced by Topoisomerase inhibitor CPT as in Aze et al 2016).

[RESPONSE] We have followed the reviewer's suggestion and detected RPA foci in addition to the pATR foci pattern included in our original manuscript in WT and DNA2 mutant cells. We have now also included aphidicolin (APH)-treated WT cells as a negative control. We observed accumulation of RPA and pATR foci at the centromeres in DNA2 null cells but not in the WT cells with or without aphidicolin treatment (Figure 6C and 6D). The new data have further strengthened our conclusion of ATR hyperactivation at the centromere in DNA2 null cells. We have included the additional data and explanations in the revised manuscript (Figure 6C and 6D and Page 14).

-The CPT experiment presented in Figure EV6A is not described in the text.

-If the authors want to compare the effects of CTP and DNA2 on ATR activation, they need to show that DNA2 loss and CPT have a similar mechanism of action (e.g. Is pATR activation by CPT peaking at centromeres? Is CPT inducing RPA accumulation? Do CPT and Aphidicolin have different effects on centromere pATR activation?).

[RESPONSE] We originally included the CPT experiment as technical control for immunostaining of phospho-ATR in the previous submission. The CPT effect on ATR activation is not our focus. Due to the limited extended views that are allowed to be included, we have removed the original Figure S6.

Minor corrections

-Aze et al showed that ATR Inhibition and not activation is important for centromere replication. Therefore, it is better to substitute the sentence on Page 13 "ATR activation is important for ensuring a sufficient time for DNA replication machinery to deal with centromeric replication problem (Aze et al, 2016)" with "Modulation of ATR activity is important for ensuring a sufficient time for DNA replication machinery to deal with centromeric replication problem (Aze et al, 2016).

[RESPONSE] We have substituted the original sentence with "Thus, modulation of ATR activation activity is important for ensuring sufficient time for the DNA replication machinery to replicate the centromere (Aze et al, 2016)." (Page 15).

-On page 14 it should be "we modified the SMARD technology to specifically" as SMARD has been already developed for telomere replication detection.

[RESPONSE] We have modified the statement as suggested (Page 16).

-On page 13 "Costanzo et al 2016" should be substituted with Aze et al 2016

[RESPONSE] "Costanzo et al 2016" has been substituted with Aze et al 2016 (Page 14).

-On page 16 the authors should properly cite the existing literature. They write: "A recent studies observed extended ssDNA regions at centromeric DNA regions, especially under external stresses (Aze et al, 2016). The origin and function of these extended ssDNA regions are not clear"

The sentence should be rewritten and should incorporate the concepts described below. Centromere intrinsic topological structure kept together by CENPA and SMC2-4 prevents extensive ssDNA accumulation in unchallenged conditions. Centromere were shown to accumulate ssDNA in response to fork stalling only when the centromere structure was disrupted.

[RESPONSE] We have rewritten the statement accordingly (Page 18).

Title: I think it should be revised. I suggest some alternative titles: "hDNA2 promotes centromeric DNA replication and genome stability"; "hDNA2 ensures genome stability by promoting centromeric DNA replication"

[RESPONSE] We have revised the title to "hDNA2 promotes centromeric DNA replication and genome stability" (Page 1).

For the discussion the authors could mention that several repair proteins were shown to accumulate on centromeric DNA, which might also be important for centromeric DNA stability and replication.

[RESPONSE] In the Discussion, we now include the following statement: Indeed, the DNA secondary structure within centromeric DNA mimics replication challenges induced by replication stressing reagents and may require involvement of DNA repair factors such as MSH2-6, XRCC5, MUS81, XRCC1 and RAD50 (Aze et al, 2016) (Page 16).

Referee #3

In the extensively revised manuscript Li and colleagues provide additional evidence that suggests that DNA2 plays a role in centromeric replication. Experiments performed as suggested by the reviewers and the development of the SMARD technique has significantly enhanced support of the conclusions of the manuscript. All previous concerns of this reviewer has been satisfactorily addressed in the revised manuscript.

[RESPONSE] We thank the reviewer for the positive feedback.

3rd Editorial Decision

10 April 2018

Thank you again for submitting your re-revised manuscript on DNA2 for our consideration. The original referees 1 and 2 have now looked at it once more, and save for a minor amendment requested by referee 2 have no further objections against publication.

There remain however several important editorial issues, and I am therefore returning the study to you once more in order to take care of these various.

REFeree REPORTS

Referee #1:

The authors properly addressed all the previous comments of this reviewer. The revised version of the manuscript is significantly improved and is now acceptable for publication in The EMBO Journal.

Referee #2:

The authors have done a great job. The new additions and clarifications address all my previous concerns.

Minor point

At page 18 after the sentence:

"centromeres accumulate ssDNA in response to DNA replication fork stalling"

the following should be added: "when centromeric DNA organization is disrupted".

3rd Revision - authors' response

16 April 2018

Thank you very much for your letter on April 10th, 2018 regarding the editorial issues with our manuscript. We have now provided all of the additional documents as you requested, including the separated production-quality PDF figure files, a "Data Accessibility" paragraph, a synopsis with the "bullet points", updated author checklist, as well as the answers to the data editor's comments and inquiries in the manuscript text.

YOU MUST COMPLETE ALL CELLS WITH A PINK BACKGROUND ↓
PLEASE NOTE THAT THIS CHECKLIST WILL BE PUBLISHED ALONGSIDE YOUR PAPER

Corresponding Author Name: Binghui Shen
Journal Submitted to: EMBO Journal
Manuscript Number: EMBOJ-2017-96729R